# Fluid shear stress activates YAP1 to promote cancer cell motility

Hyun Jung Lee[1,2], Miguel F. Diaz[1,2], Katherine M. Price[3], Joyce A. Ozuna[3], Songlin Zhang[4], Eva M. Sevick-Muraca[5], John P. Hagan[6] & Pamela L. Wenzel[1,2]

Mechanical stress is pervasive in egress routes of malignancy, yet the intrinsic effects of force on tumour cells remain poorly understood. Here, we demonstrate that frictional force characteristic of flow in the lymphatics stimulates YAP1 to drive cancer cell migration; whereas intensities of fluid wall shear stress (WSS) typical of venous or arterial flow inhibit taxis. YAP1, but not TAZ, is strictly required for WSS-enhanced cell movement, as blockade of *YAP1*, *TEAD1-4* or the YAP1–TEAD interaction reduces cellular velocity to levels observed without flow. Silencing of TEAD phenocopies loss of YAP1, implicating transcriptional transactivation function in mediating force-enhanced cell migration. WSS dictates expression of a network of YAP1 effectors with executive roles in invasion, chemotaxis and adhesion downstream of the ROCK–LIMK–cofilin signalling axis. Altogether, these data implicate YAP1 as a fluid mechanosensor that functions to regulate genes that promote metastasis.

[1] Children's Regenerative Medicine Program, Department of Pediatric Surgery, University of Texas Health Science Center at Houston, Houston, Texas 77030, USA. [2] Center for Stem Cell and Regenerative Medicine, The Brown Foundation Institute of Molecular Medicine, University of Texas Health Science Center at Houston, Houston, Texas 77030, USA. [3] Department of BioSciences, Rice University, Houston, Texas 77030, USA. [4] Department of Pathology, The University of Texas Medical School, Houston, Texas 77030, USA. [5] Center for Molecular Imaging, The Brown Foundation Institute of Molecular Medicine, University of Texas Health Science Center at Houston, Houston, Texas 77030, USA. [6] Vivian L. Smith Department of Neurosurgery, University of Texas Health Science Center at Houston, Houston, Texas 77030, USA. Correspondence and requests for materials should be addressed to P.L.W. (email: Pamela.L.Wenzel@uth.tmc.edu).

Biophysical cues in the microenvironment such as stiffness of the extracellular matrix, nanotopography and biomechanical force have gained significant attention in recent years for their roles in defining fundamental cell properties, including cell fate, self-renewal, motility and homing behaviours[1]. Mechanical features of the tumour microenvironment are altered by changes in tissue architecture and density, cellular composition, extracellular matrix deposition, immune cell infiltration, presence of microvasculature, and interstitial fluid flow and pressure. Most metastatic cancers initially spread from the primary tumour through the lymphatic system, a vascular network that drains interstitial tissue fluid into regional lymph node basins. Fluid frictional force or wall shear stress (WSS), is pervasive in egress routes from solid tumours and influences cytokine production and immune cell adhesion in lymphatic and venous vasculatures[2]. Flow in and around solid tumours influences extracellular gradients of growth factors and chemokines, transport of tumour antigens and delivery of chemotherapeutic agents[3,4] but the impact of flow-associated biomechanical force on intrinsic tumour cell biology and malignancy remains poorly understood[5].

Yes-associated protein 1 (YAP1) and its paralog, transcriptional coactivator with PDZ-binding motif (TAZ), were recently shown to be exquisitely sensitive to matrix stiffness, cell density and shape, and mechanical stretching[6–8]. YAP1 and TAZ in active form translocate to the nucleus and associate with the TEAD family of transcription factors to regulate cell proliferation, tissue growth and differentiation[9,10]. In human breast, ovarian, liver and colorectal cancers, YAP1/TAZ activity positively correlates with chemoresistance, frequency of self-renewing cancer stem cells, tumour heterogeneity, histological grade and metastasis[11–13]. The chromosome region 11q22 containing YAP1 is amplified in several human tumours; however, in the absence of 11q22 amplification, it is not well understood how YAP1 and TAZ may be dysregulated in cancer[14,15]. YAP1 and TAZ appear to respond to several upstream regulatory inputs[16] but, to date, no reports have directly demonstrated regulation of YAP1 or TAZ by mechanical cues in the context of tumour biology.

Here, we demonstrate that WSS characteristic of flow within the lymphatic vasculature regulates YAP1/TAZ to modify cancer cell motility using a soft-polymer microfluidics system engineered for the study of mechanobiology. Lymph node metastases from orthotopic xenografts express YAP1 more strongly than primary tumours. Inhibition of YAP1, but not TAZ, by siRNA, inhibitory peptide or pharmacological disruption of the YAP1–TEAD interaction results in significantly reduced WSS-induced motility. Silencing of TEAD phenocopies loss of YAP1, implicating transcriptional transactivation function in mediating force-enhanced cell migration. A combination of shRNA-based knockdown and pharmacological inhibition implicates Rho kinase (ROCK), LIM-domain kinase (LIMK) and cofilin upstream of YAP1 in transduction of flow-based mechanical cues. This work suggests YAP1 and TAZ play distinct roles in the response to mechanical cues and identifies the ROCK–LIMK–YAP1 signalling axis as a central component of the mechanosensory transduction machinery that promotes flow-induced motility of cancer cells.

## Results

**Fluid shear stress promotes motility in cancer cells.** Evidence supports that dendritic cells and most probably cancer cells, enter the lymphatic vasculature at sites of interstitial fluid uptake[4,17]. These areas of initial lymphatics and immature lymphatics are formed through the process of tumour lymphangiogenesis, and consist of blind-ended endothelial structures with wide lumina that empty into collecting lymphatics. Collecting lymphatic ducts possess unidirectional valves and contractile vascular smooth muscle cells that promote egress of interstitial fluid towards regional lymph node basins. Shear stresses within initial or immature lymphatics are estimated to be $<0.2–1$ dyne cm$^2$, whereas transport downstream in larger collecting vessels is pulsatile and can reach maximal intensities of 5 dyne cm$^{-2}$ at the vessel wall[18,19]. To evaluate specifically the effects of fluid force on cancer cells, we microengineered a biomimetic platform to model mechanical properties and predictions of fluid movement across tumour cells (Fig. 1a; Supplementary Fig. 1a). Briefly, soft lithography was used to create polydimethyl siloxane (PDMS) microchips with an elastic modulus comparable to the vascular wall[20]. The lumen of the culture surface was coated in collagen matrix, followed by seeding and 24-h adaptation of cells to soft substrate elasticity. Programmable syringe pumps managed laminar flow of medium through microfluidic channels at a constant flow rate, with unidirectional flow of fresh medium to minimize spatial cues from secreted paracrine factors.

Upon exposure to 0.05 dyne cm$^{-2}$ WSS, we observed extensive development of fine cytoplasmic extensions or filopodia, in the highly metastatic PC3 human prostate cancer cell line (Fig. 1b). MMP2 and MMP9 matrix metalloprotease expression and activity were stimulated within 1 h of WSS (Fig. 1c,d). We initially monitored motility behaviour by time-lapse imaging of cultures exposed to 6 h of WSS or static conditions (see description of time-lapse imaging in 'Methods' section). Overall cellular velocity of PC3 was increased ($0.47 \pm 0.01$ μm min$^{-1}$ under static conditions versus $0.67 \pm 0.02$ μm min$^{-1}$ under WSS; $n = 7$ independent experiments; the results are expressed as mean ± s.e.m.), consistent with response of the moderately metastatic DU145 prostate line ($0.15 \pm 0.02$ μm min$^{-1}$ static versus $0.23 \pm 0.04$ μm min$^{-1}$ WSS; $n = 3$ independent experiments; the results are expressed as mean ± s.e.m.) (Fig. 1e,f; Supplementary Movies 1 and 2). Measurement of velocity during finite intervals of time revealed that migration speed was dramatically increased within 30 min of WSS initiation and was sustained throughout the duration of the experiment (Fig. 1g). Cellular velocities were also increased at higher magnitudes of WSS, up to 5 dyne cm$^{-2}$, at which point WSS appeared to inhibit cell movement (Supplementary Fig. 1b,c). Motility was not uniformly directional as has been demonstrated in response to flow-induced CCR7 gradients[4], suggesting that increased movement was not in response to chemokine spatial cues produced by fluid flow. Taken together, these data implicated WSS in activation of intrinsic signalling pathways that drive invasive behaviour of cancer cells.

**YAP1 signalling is altered by WSS.** To define the signalling mechanisms triggered by WSS that could mediate effects on motility, we conducted global gene expression profiling of cells following 3 h of WSS or static culture. Briefly, WSS was applied to PC3 cells cultured within microfluidic channels, replicates were lysed, and RNA was processed for analysis by Illumina Human HT-12 v4.0 Expression BeadChips covering 47,231 transcripts. Differential gene expression analysis ($P < 0.01$, 2-fold threshold) revealed significant change in 439 unique transcripts (Supplementary Data 1). By ingenuity pathway analysis, differentially expressed genes were found to encode enzymes and transcription regulators required for anti-apoptotic and proliferative signalling associated with several cancers, including small cell lung, colon, pancreatic, thyroid and prostate cancer (MYC, KRAS, CTNNB1, ABL1, CDKN1B, BAD, NFKB2 and NFKBIE) (Fig. 2a). Components of the death receptor and apoptosis signalling pathways were also significantly enriched

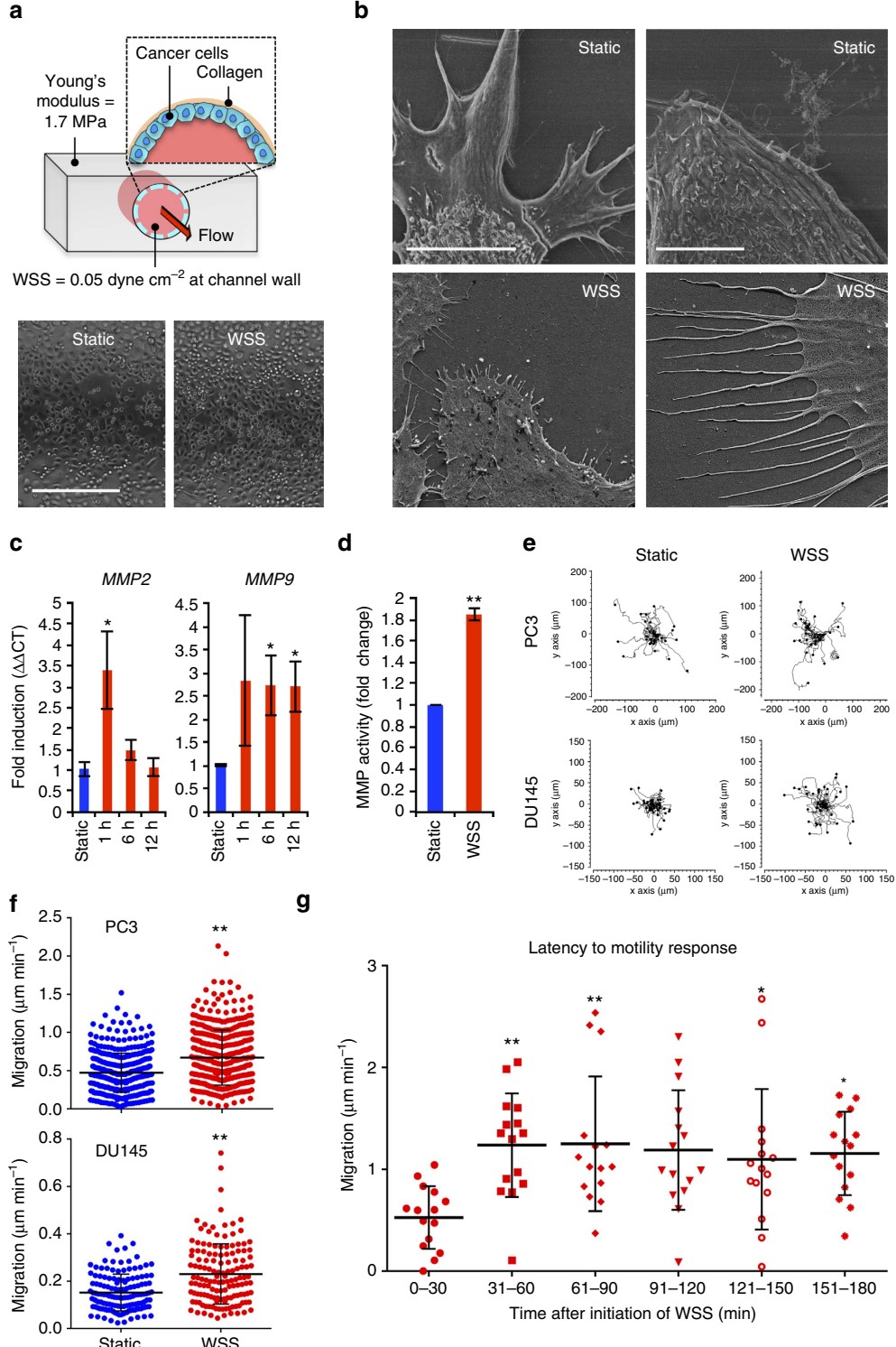

**Figure 1 | Fluid flow stimulates motility and matrix metalloprotease activity.** (**a**) Cylindrical PDMS fluidics channels coated in collagen support a monolayer of cells. Flow of media through the culture chamber exposes cells to WSS of 0.05 dyne cm$^{-2}$. Scale bar on bright field photomicrograph of PC3 cells within the scaffold represents 400 µm. (**b**) Filopodia formation in response to WSS is extensive. Scale bar in left panel, 10 µm, scale bar in right panel, 5 µm. (**c**) Transcription of *MMP2* and *MMP9* is stimulated by WSS ($n = 3$ independent experiments; Kruskal–Wallis one-way ANOVA, $P < 0.001$). (**d**) Total MMP activity measured by fluorogenic peptide substrate digestion assays was increased by exposure to 6 h WSS ($n = 3$ independent experiments; unpaired $t$-test, **$P < 0.0001$). (**e**) Spatial tracking of PC3 and DU145 cancer cell movement during 6 h of time-lapse imaging, where each cell lies at the origin (0,0) at $t = 0$ h. Plots depict motility of individual cells in one representative experiment. (**f**) Quantification of migration speed reveals increased cellular velocities of individual cells under WSS. ($n = 7$ independent experiments, two-tailed $t$-test, **$P = 4.22E − 18$ for PC3 cells; $n = 3$ independent experiments, two-tailed $t$-test, **$P = 1.56E − 9$ for DU145 cells). (**g**) Time segmented migration speed of cells after WSS initiation (Kruskal–Wallis one-way ANOVA, *$P < 0.05$, **$P < 0.01$). Error bars represent ± s.e.m.

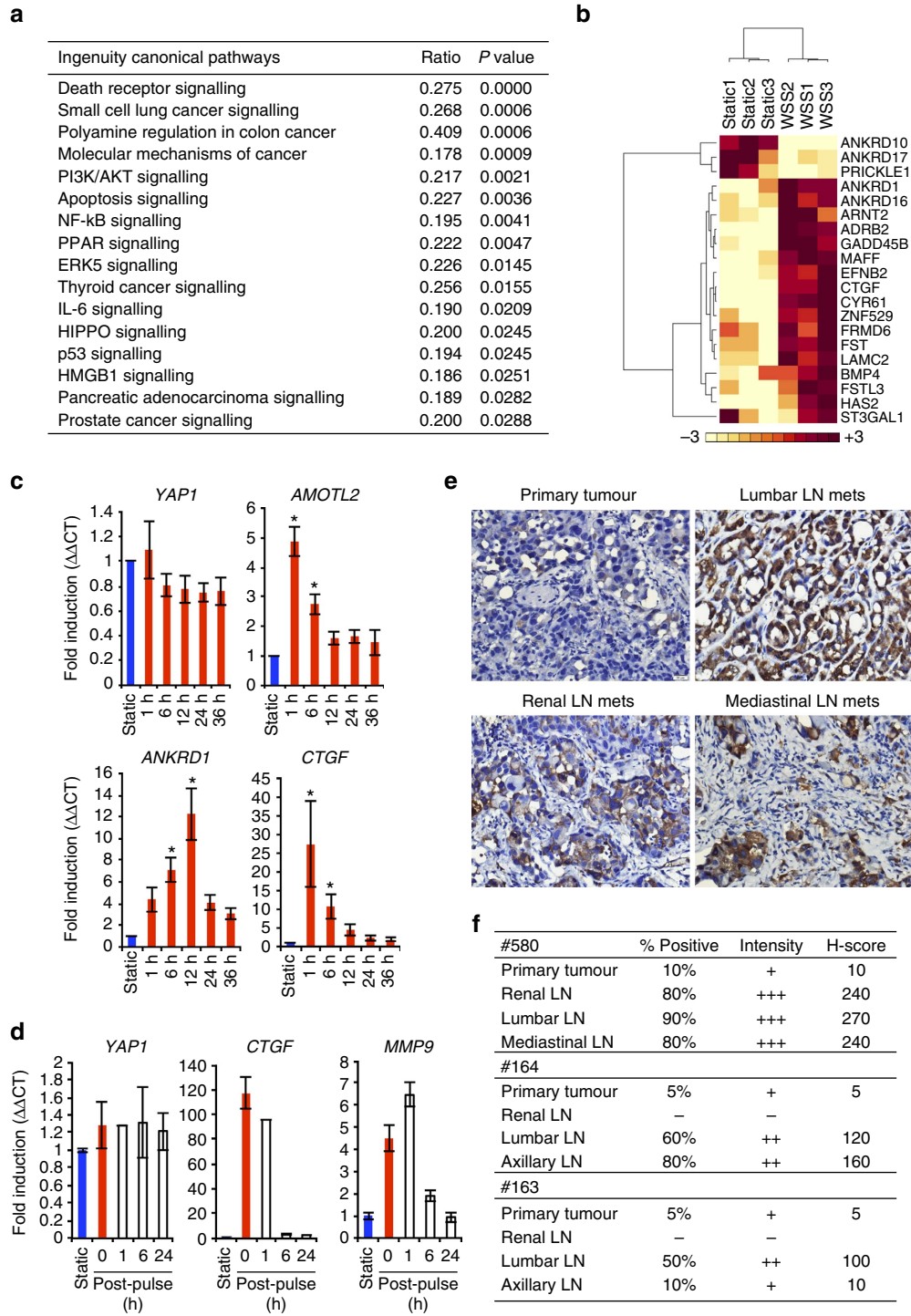

**Figure 2 | YAP1 signalling is modulated by fluid WSS and tumour location.** Gene expression was determined by Illumina HT-12 BeadChips in PC3 cells exposed to WSS or static conditions for 3 h. (**a**) Ingenuity pathway analysis reveals roles for WSS in a number of canonical pathways. (**b**) Hierarchical clustering of genes regulated by YAP1 segregates static and WSS-exposed cells ($P < 0.05$). Colour key represents normalized expression within rows. (**c**) Elevated expression of YAP1/TAZ target genes was validated at several time points following initiation of WSS by qRT-PCR ($n = 3$ independent experiments; Kruskal–Wallis one-way ANOVA, $P < 0.001$, pairwise group differences were compared using the Dunn's test, static versus WSS individual time point comparison, $*P < 0.05$). (**d**) YAP1/TAZ and target genes expression were evaluated at static, WSS for 1 h and each recovery time point after WSS stopped. (**e**) Immunohistochemical staining of YAP1 in primary tumour and lymph node metastases (LN mets) of representative mouse implanted orthotopically with PC3 cells. Scale bar, 20 μm. (**f**) H-score of YAP1 protein expression in primary tumors and associated lymph nodes for three individual animals was calculated as the product of YAP1 intensity values (1–3) and per cent positive PC3 cells in the tumour (0–100), ranging from 0 to 300. Error bars represent ± s.e.m.

(*ACTG2*, *BIRC3*, *CASP8* and *CHUK*). A number of other kinases, enzymes and transcription factors with recognized roles in prostate carcinogenesis and cancer progression were found to contribute to signalling through PI3K/AKT, PPAR, ERK5, IL-6 and HMGB1. Notably, phosphatases (*PPP1R3C* and *PPP2CA*), transcription regulators (*WWC1* and *SMAD3*) and signal transduction factors (*YWHAB* and *YWHAG*) reported to be components of the HIPPO signalling pathway in the ingenuity knowledge base were differentially expressed ($P = 0.02$) (Fig. 2a; Supplementary Fig. 2a). Altered expression of genes involved in HIPPO signalling prompted us to conduct *in silico* examination of YAP1 target genes. We included only YAP1-responsive genes shown to represent *bona fide* targets by overlap in promoter occupancy via ChIP-seq and altered gene expression resulting from gain or loss of function studies in cardiomyocytes and SF268 glioblastoma cells[21,22] (Supplementary Data 2). Of 83 previously identified YAP1 targets, we found that 3 h WSS exposure significantly upregulated 17 genes and downregulated 3 genes (Fig. 2b). When evaluated against a recent report of 379 genes regulated by enhancers occupied by YAP1/TAZ/TEAD in MDA-MB-231 breast cancer cells[23], we found significant upregulation in 106 distinct genes and downregulation of 24 genes (Supplementary Data 3). Expression of a subset of these genes and others involved in motility was validated in PC3 cells at several time points following initiation of WSS, including 1, 3, 6, 12, 24 and 36 h (Fig. 2c; Supplementary Fig. 2a,b). *AMOTL2* (4.8-fold, $P < 0.05$) and *CTGF* (27-fold, $P < 0.05$) were rapidly increased within 1 h after WSS and reduced gradually, though were sustained long-term at levels higher than static conditions. *ANKRD1* (12-fold, $P < 0.05$) peaked 12 h after WSS. On termination of WSS, expression returned to levels observed in static conditions within 6 h (Fig. 2d). *CYR61* was upregulated at 3 and 6 h after WSS (2-fold, $P < 0.05$) (Supplementary Fig. 2a).

YAP1 has recently been shown to associate with adverse outcomes in human prostate cancer and, in murine xenograft models, to correlate with castration resistance and invasion[24,25]. In contrast, an independent report suggests that loss of YAP1 correlates with increased Gleason score[26]. As with many cancers, prostate cancer initially metastasizes through the lymphatics and, as a result, staging is performed through pelvic lymph node dissection at time of radical prostatectomy. To model this transition in disease progression, we established primary prostate tumours in mice orthotopically transplanted with PC3 cells and assessed YAP1 expression in the primary tumour and lymph nodes. PC3 cells stably expressing the DsRed-Express fluorescent protein gene reporter were implanted in mice to enable longitudinal imaging of metastasis from the prostate and, at sacrifice, provided guidance for resection and an assessment validated through pathological examination of lymph node cancer status, as described previously[27]. Approximately 50% of the animals implanted with PC3-DsRed-Express cells presented with lymph node metastasis as assessed from fluorescence imaging. Of mice found to express DsRed-Express in the lymph nodes, tumour burden was verified histologically by H&E in the prostate of 4 of 4 mice, and 3 of these were confirmed to have lymph node metastases. Primary tumours expressed YAP1 in 5–10% of cells that comprised the tumour; whereas, 10% as lowest and up to 90% of the tumour cells in the lymph nodes expressed YAP1 (Fig. 2e,f). Corresponding H&E images are included in Supplementary Fig. 3. This enrichment in the number of cells expressing YAP1 in the lymph nodes was accompanied by significant increase in YAP1 staining intensity, indicating that PC3 cells in the lymph nodes express higher levels of YAP1 protein. Altogether, these data showed that YAP1 expression in the primary tumour is low despite invasive potential of the cells, and suggested that YAP1/TAZ signalling could be activated by biophysical cues present in the lymphatics and/or lymph nodes to contribute to high YAP1 expression in metastases.

**Distinctive roles of YAP1 and TAZ downstream of WSS.** To identify specifically the contribution of force to regulation of YAP1 and TAZ, we analysed protein expression and subcellular localization following culture under static or WSS conditions. Within 6 h of WSS, YAP1 and TAZ translocated to the nucleus (Fig. 3a). When quantified, nuclear YAP1 and TAZ colocalized and were detectable in a higher percentage of cells following WSS (51.7 ± 3.3% static versus 77.7 ± 2.9% WSS) (Fig. 3b). The HIPPO pathway engages LATS1/2 kinase to phosphorylate and inactivate YAP1 in response to various stimuli, including energetic stress and actin depolymerization, and is itself activated by phosphorylation[28–30]. LATS kinase inactivates YAP by phosphorylation of at least five serine residues, including S61, S109, S127, S164 and S381 (ref. 31). Phosphorylation of S127 promotes YAP1 interaction with 14-3-3 and sequestration of YAP1 in the cytoplasm. We found that WSS reduced phosphorylation of S127, consistent with increased nuclear localization observed in immunofluorescent staining (Fig. 3c). Total TAZ protein level was also slightly elevated, suggesting that YAP1 and TAZ activity are both stimulated by WSS in prostate cancer cells. Changes in protein localization induced by WSS were accompanied by activation of a fluorescent reporter of YAP1/TAZ activity containing TEAD enhancer elements (p8xGTIIC-DsRed-Monomer). Reporter activity was detectable as early as 120 min after WSS initiation and was sustained throughout duration of exposure to flow, suggesting that WSS is a potent regulator of YAP1/TAZ transactivation function (Fig. 3d; Supplementary Movies 3 and 4).

Constitutively active forms of YAP1 have been shown to stimulate cancer cell migration and invasion[10]. We therefore tested the requirement for YAP1 and TAZ in motility induced by WSS using siRNA-based gene silencing. *YAP1* knockdown attenuated cellular motility induced by WSS, whereas *TAZ* knockdown had no effect on taxis (control siRNA, 0.73 ± 0.09 μm min$^{-1}$; *YAP1* siRNA, 0.44 ± 0.07 μm min$^{-1}$; *TAZ* siRNA, 0.77 ± 0.01 μm min$^{-1}$; $n = 3$ independent experiments; the results are expressed as mean ± s.e.m.) (Fig. 3e–g). Despite the failure of *TAZ* siRNA to interrupt motility, our data was consistent with TAZ activation by WSS, as knockdown of *TAZ* suppressed WSS-induced increases in *ANKRD1* and *CTGF* (Supplementary Fig. 4a). YAP1 knockdown did not reduce WSS-induced *ANKRD1* or *CTGF* transcript levels. Not unexpectedly, knockdown of *CTGF* failed to reduce cellular velocity (Fig. 3h,i). Overexpression of wild-type TAZ or the constitutively active TAZ S89A was insufficient to rescue motility in the absence of YAP1 (Supplementary Fig. 4b). Altogether, these data suggest that YAP1 and TAZ contribute unique functions in mechanotransduction and that WSS-induced motility of prostate tumour cells requires YAP1.

**ROCK–LIMK–cofilin axis regulates YAP-mediated motility.** We observed that WSS enriches for central F-actin compared with peripheral F-actin in PC3 cells (Supplementary Fig. 5a). Recently, Aragona *et al.*[8] showed that the actin filament modulators cofilin and gelsolin regulate YAP1 and TAZ function. Cofilin and gelsolin proteins are F-actin-severing proteins that play an essential role in cytoskeletal rearrangement. We confirmed expression of cofilin and gelsolin in PC3 cells and generated stable cell lines expressing cofilin and gelsolin shRNA to test their contribution to WSS-induced motility (Fig. 4a). Knockdown of cofilin significantly increased cellular velocities in static and WSS conditions, while silencing of

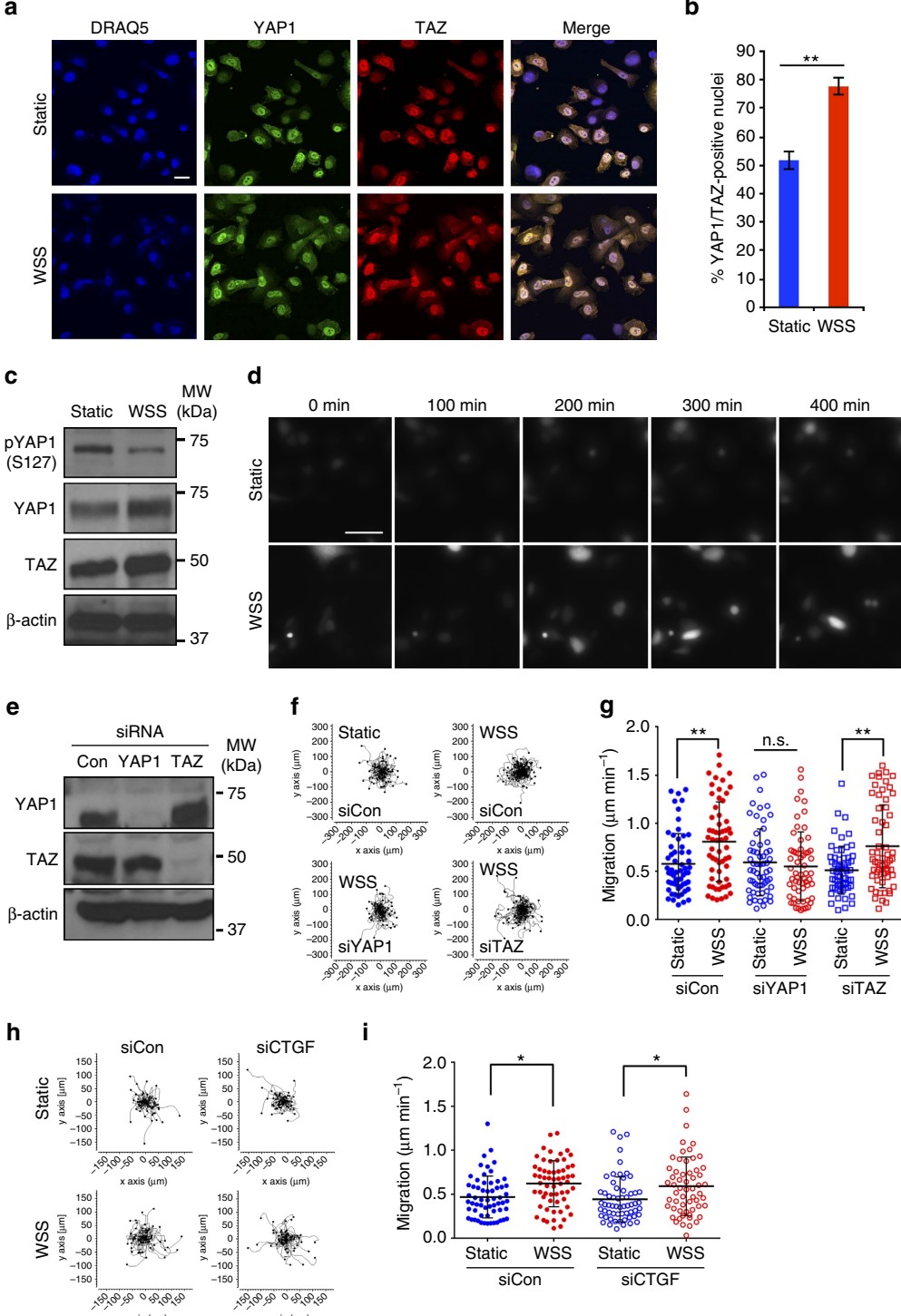

**Figure 3 | YAP1 drives flow-induced cellular motility.** (**a**) YAP1 and TAZ localize to the nucleus in response to WSS (6 h). Nuclei were detected by Draq5. Scale bar is equivalent to 25 µm (n = 3 independent experiments). Three or more images were taken in each experiment and representative images are shown. (**b**) The frequency of YAP1/TAZ nuclear localization significantly increases with fluid flow (n = 3 independent experiments; two-tailed t-test, **P < 0.0001). (**c**) Consistent with translocation to the nucleus, WSS stimulated dephosphorylation of serine-127 YAP1 and slightly elevated TAZ expression. (**d**) Still photos from time-lapse DsRed movie file for the 8xGTIIC-DsRed-Monomer TEAD reporter positive cells with static or WSS. Scale bar, 50 µm. (**e**) Efficiency of YAP1 and TAZ knockdown was measured by western blot analysis of PC3 cells transfected with control siRNA or siRNAs against YAP1 or TAZ. (**f**) Time-lapse imaging was used to measure motility during 6 h of WSS or static culture. (**g**) Quantification of migration speed of individual cells shows that knockdown of YAP1, but not TAZ, impairs motility stimulated by WSS (n = 3 independent experiment, Kruskal–Wallis one-way ANOVA, **P < 0.001). (**h**) Motility plots from time-lapse images when YAP1/TAZ downstream target, AMOTL2 or CTGF, was absent. (**i**) AMOTL2 and CTGF knockdown did not affect enhanced migration speed induced by WSS. Error bars represent ± s.e.m.

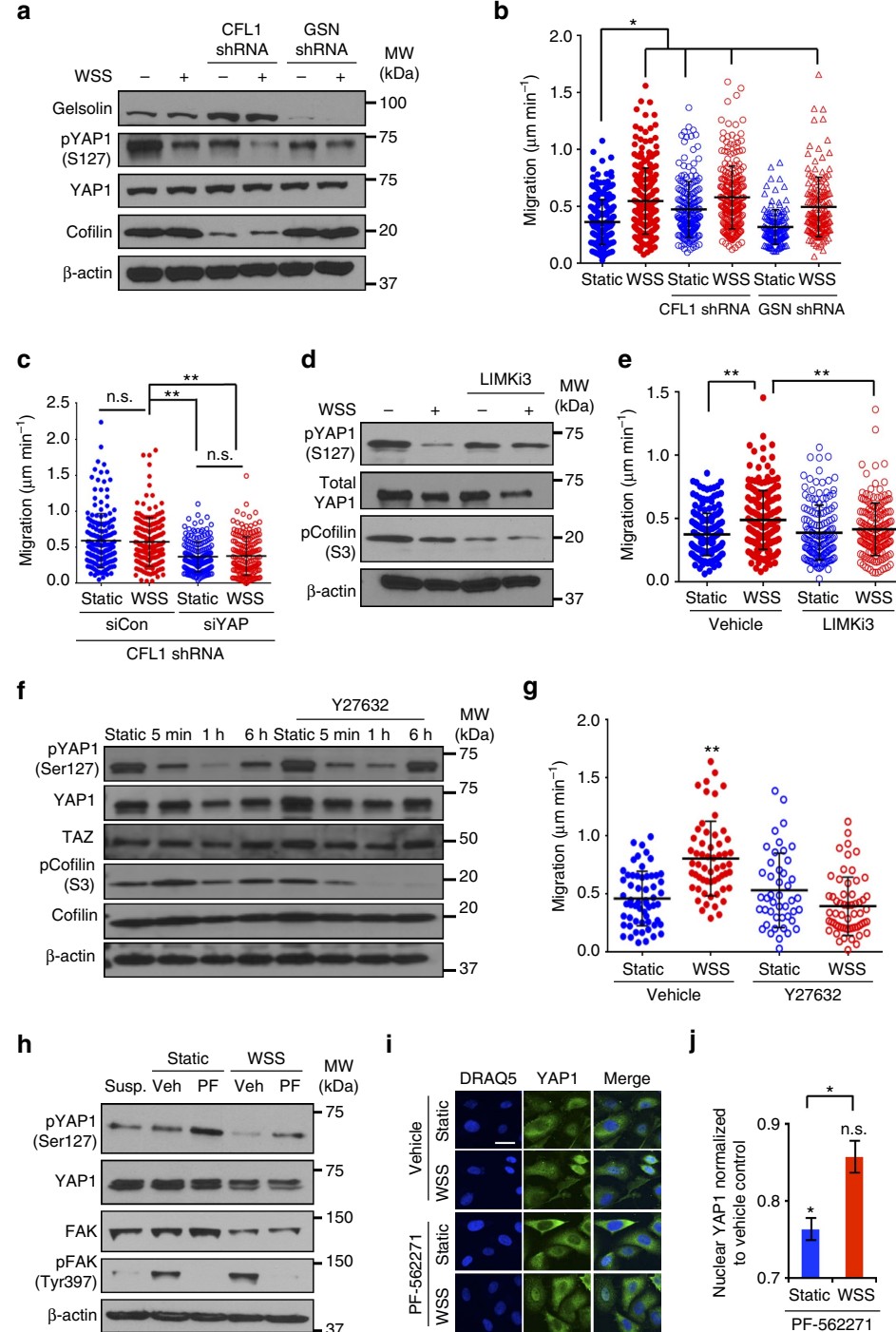

**Figure 4 | ROCK–LIMK–cofilin axis is required for WSS-induced YAP1 activity.** (**a**) Cofilin, gelsolin and YAP1 protein levels were measured in stable PC3 lines generated by lentiviral shRNA. (**b**) Gelsolin knockdown did not significantly change movement speed, but silencing of cofilin increased velocities of cells cultured under WSS as well as static conditions ($n = 3$ independent experiments, Kruskal–Wallis One-way ANOVA, **$P < 0.001$; multiple comparison of individual group was tested by Dunn's method, *$P < 0.05$). (**c**) Enhanced motility in cofilin knockdown cells in static and WSS conditions was significantly reduced by YAP ablation ($n = 3$ independent experiments, Kruskal–Wallis One-Way ANOVA, **$P < 0.001$). (**d,e**) LIMK inhibitor (LIMKi3) elevates levels of phosphorylated YAP1 and blocks motility ($n = 3$, Kruskal–Wallis One-way ANOVA, **$P < 0.001$). (**f**) Treatment with the ROCK inhibitor Y27632 elevated levels of phosphorylated S127 YAP1 present in WSS-treated PC3 cells, suggesting that ROCK normally plays an important role in promoting dephosphorylation of YAP1 to interrupt its sequestration in the cytoplasm. (**g**) WSS-induced motility is blocked by inhibition of ROCK (unpaired $t$-test, **$P < 0.0001$). (**h**) WSS increases FAK phosphorylation at Y397 in PC3 cells at 30 min. The PF-562271 FAK inhibitor reduces phosphorylation of FAK and increases phosphorylation of YAP1. (**i**) Representative images of YAP1 subcellular localization at 30 min of static or WSS culture. Cells in suspension express negligible amounts of pFAK. (**j**) Inhibition of FAK reduces YAP1 nuclear localization in static cultures (unpaired $t$-test, **$P = 0.002$) and, to a lesser extent, in WSS cultures (unpaired $t$-test, $P = 0.07$). Decrease in YAP1 nuclear localization differed significantly between static and WSS when normalized to vehicle controls (unpaired $t$-test, *$P = 0.02$), suggesting that WSS can act via FAK-independent mechanisms to stimulate YAP1. Error bars represent ± s.e.m.

gelsolin had no significant effect on migration speed (Fig. 4b). Moreover, ablation of YAP1 diminished elevated velocity of cofilin knockdown cells in both static and WSS conditions (Fig. 4c), in support of the notion that enhanced motility of cofilin shRNA-expressing cells is mediated by YAP1. Cofilin causes depolymerization at the minus end of actin filaments, thereby preventing reassembly. LIM-domain kinase (LIMK) inactivates cofilin through phosphorylation[32] and thus is expected to stabilize F-actin. We found that blockade of LIMK by the potent inhibitor LIMKi3 led to accumulation of dephosphorylated (active) cofilin and inactive phospho-S127 YAP1 (Fig. 4d). Importantly, LIMKi3 treatment also reduced motility in WSS-exposed cultures (Fig. 4e; Supplementary Fig. 5b). This result suggests that LIMK inhibition activates cofilin, thereby accelerating actin depolymerization, attenuating YAP1 function and limiting cellular motility. Taken together, we demonstrate that cofilin negatively regulates YAP1 activity to control cellular motility, likely via actin filament remodelling.

WSS has previously been shown to stimulate Rho kinase (ROCK)[33] and the MAPK kinase family, including extracellular signal-regulatory kinase (ERK) and c-Jun NH2-terminal kinase (JNK)[34,35]. As LIMK can be regulated by the ROCK and ERK pathways[36,37], we hypothesized that ROCK and ERK may lie downstream of WSS in regulation of YAP1. We first examined the effects of ROCK and ERK inhibition on the activity of cofilin and YAP1. Following initiation of WSS, cofilin is rapidly phosphorylated (inactivated). Cofilin is modestly reactivated at 1 h, but is again inactivated by 6 h of WSS (Fig. 4f). In the presence of ROCK inhibitor, Y27632, cofilin undergoes dephosphorylation within 5 min of WSS, resulting in an active state for the remaining duration of the WSS exposure to 6 h. This disruption in cofilin phosphorylation dynamics supports a role for ROCK as a key upstream kinase in LIMK-mediated phosphorylation of cofilin. Importantly, ROCK inhibition via Y27632 treatment increased phosphorylated YAP1 (S127) and decreased migration speed in the presence of WSS (Fig. 4f,g; Supplementary Fig. 5c).

Focal adhesion kinase (FAK) regulates Rho activity downstream of integrin signalling to modulate motility[38]. We therefore evaluated whether FAK mediates activation of YAP1 by WSS. On rigid, collagen-coated tissue culture plastic (not PDMS), autophosphorylation of FAK at Y397 was robustly elevated by WSS, an indicator that the first autophosphorylation step of FAK activation was initiated (Supplementary Fig. 5d). In contrast, cells cultured briefly in suspension express very low levels of pFAK Y397 (Fig. 4h; Supplementary Fig. 5d). On soft PDMS surfaces used in our biomimetic platform, Y397 phosphorylation of FAK was increased, albeit somewhat attenuated relative to the rigid surface (Fig. 4h). Total FAK also appeared to be reduced by WSS on PDMS, but not on plastic (Fig. 4h; Supplementary Fig. 5d). Constitutively high levels of β1-integrin activation contribute to elevated FAK activity in PC3 cells and prostate cancer metastases[39]. We therefore inhibited FAK with a potent and selective inhibitor (PF-562271) shown previously to effectively suppress FAK activation in PC3 cells[40]. In static cultures, FAK inhibition increased levels of phosphorylated YAP1 and reduced YAP1 nuclear localization (unpaired $t$-test, $P = 0.002$) (Fig. 4h–j). After 30 min of WSS, FAK inhibition reduced YAP1 nuclear localization, albeit at more modest and non-significant levels (unpaired $t$-test, $P = 0.07$). Indeed, exclusion of YAP1 from the nucleus by FAK inhibition was significantly more pronounced in static versus WSS cultures (unpaired $t$-test, $P = 0.02$). It is possible that incomplete inhibition of FAK by PF-562271 could allow WSS to partially stimulate YAP1 nuclear localization; however, expression of pFAK Y397 following PF-562271 treatment resembled cells in suspension, which contain

negligible levels of FAK activity. Alternatively, WSS may activate YAP1 via FAK-independent mechanisms, consistent with previous studies suggesting that focal adhesions are not required for YAP1 nuclear localization[7,41].

Application of an ERK inhibitor (U0126) delayed cofilin dephosphorylation by WSS, but was unable to block completely the response to mechanical stress (Supplementary Fig. 5e,f). Cellular motility was significantly impacted by ERK inhibition, suggesting that the ERK pathway is triggered by WSS and is required for motility response, but is not responsible for modulation of the LIMK–cofilin–YAP1 signalling axis.

**YAP1–TEAD interaction is required for WSS-induced motility**. A constitutively active form of YAP1 containing an alanine substitution at S127 (S127A) has previously been shown to stimulate motility and promote metastasis from breast cancer and melanoma xenografts in mice[10]. Consistent with these findings, YAP1 S127A, but not the activated 5SA/S94A mutant incapable of binding TEAD transcription factors, significantly elevated cellular velocity of PC3 and LNCaP prostate cancer cells cultured under static conditions (Fig. 5a–c; Supplementary Fig. 6a). Application of a small molecule inhibitor of the YAP1–TEAD interaction, verteporfin[42] and YAP1–TEAD inhibitory peptide (YTIP)[43] confirmed that association of YAP1 and TEAD is required for WSS-induced motility (Fig. 5d; Supplementary Fig. 6b,c). *AMOTL2*, *ANRKD1* and *CTGF* expression levels were induced by WSS when verteporfin was present in the medium, confirming that YAP1-dependent motility is not mediated by these genes (Supplementary Fig. 7).

To evaluate which of the TEAD family members may be critical for YAP1-driven motility, cells were transfected with siRNAs against *TEAD1*, *TEAD2*, *TEAD3* or *TEAD4* singly or in combination. Overall cellular motility in the presence of WSS was significantly reduced with single TEAD targeting, although cellular velocities remained elevated above static culture (Fig. 5e). As knockdown of YAP1 restored motility of WSS cultures to static levels (Fig. 3g), we hypothesized that functional redundancy of TEAD1-4 might support partial YAP1 response to WSS. Indeed, knockdown of all four TEAD members reduced cellular motility to the same level as static conditions ($0.37 \pm 0.05 \, \mu m \, min^{-1}$ siCon Static versus $0.42 \pm 0.08 \, \mu m \, min^{-1}$ siTEAD1-4 WSS; $n = 3$ independent experiments, Dunn's method, $P < 0.05$; the results are expressed as mean $\pm$ s.e.m.) (Fig. 5f). Notably, preservation of TEAD1 in combination knockdown led to elevated migration speed, whereas preservation of TEAD2, TEAD3 or TEAD4 did not enhance cellular motility, suggesting that TEAD1 may contribute critically to WSS responsiveness ($0.72 \pm 0.09 \, \mu m \, min^{-1}$ siCon WSS versus $0.71 \pm 0.10 \, \mu m \, min^{-1}$ TEAD1 nontarget WSS; $n = 3$ independent experiments; the results are expressed as mean $\pm$ s.e.m.) (Fig. 5f). As shown in Fig. 5e, however, knockdown of TEAD1 alone was insufficient to abolish completely response to WSS, implicating TEAD2-4 in compensation of unique functions TEAD1 may normally possess.

**Master regulators linked to YAP1-mediated motility**. Global gene expression profiling following knockdown of *YAP1* identified 73 distinct genes changed 2-fold by both WSS and loss of *YAP1* (Supplementary Fig. 8a; Supplementary Data 4). Of these genes, 23 were inversely related. A subset of those genes was upregulated by WSS and downregulated by silencing of *YAP1* (Supplementary Fig. 8a). YAP1-dependent regulation by WSS was confirmed by qRT-PCR in *KRT80* and *PLOD2*; whereas, *CTGF* transcript levels were unaffected by *YAP1* ablation (Supplementary Fig. 8b). Notably, *KRT80* and *PLOD2* both

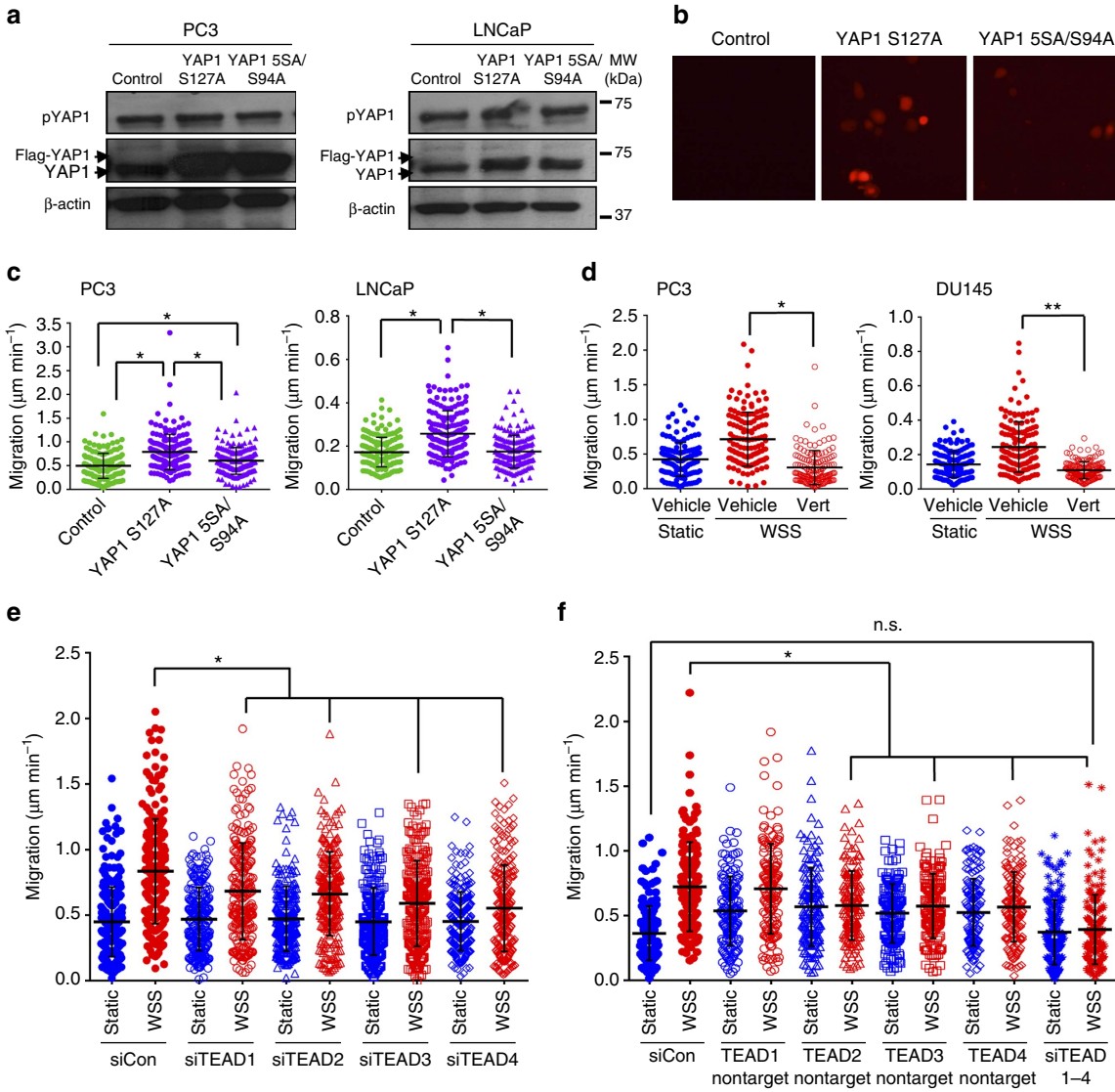

**Figure 5 | YAP1–TEAD interaction is essential for WSS-induced motility.** (**a**) YAP1 protein expression in PC3 and LNCaP cells following transfection with a constitutively active form of YAP1 (S127A) or a TEAD-binding domain mutant (5SA/S94A). (**b**) Positivity of DsRed TEAD reporter in pEGFP-N1, YAP1 S127A and YAP1 5SA/S94A transfected PC3 cells indicates low to no activation by control and TEAD-binding mutant. (**c**) YAP1 S127A stimulated cell migration, whereas YAP1 5SA/S94A failed to increase movement relative to the pEGFP-N1 (Kruskal–Wallis one-way ANOVA, **$P < 0.001$). (**d**) Quantification of motility with the YAP1–TEAD inhibitor verteporfin shows significant reduction in motility ($n = 3$ independent experiments, Kruskal–Wallis one-way ANOVA, **$P < 0.001$). (**e**) WSS-induced motility of PC3 cells was reduced with knockdown of individual TEADs (three independent experiments, all pairwise multiple comparison procedure by Dunn's method, *$P < 0.05$). (**f**) Knockdown of TEAD in combination shows reduction of motility to static levels with silencing of TEAD1-4 (three independent experiments, All pairwise multiple comparison procedure by Dunn's method, *$P < 0.05$). Error bars represent ± s.e.m.

influence assembly of major structural fibres and ECM organization. The lysyl collagenase encoded by *PLOD2* enhances collagen secretion and collagen fibril organization to enhance ECM stiffness in breast cancer[44], and *PLOD2* knockdown has been shown to reduce tumour invasiveness and metastasis to lung and lymph nodes[45]. Reduced stringency of analysis to 1.25-fold changes revealed a network of YAP1-responsive WSS signalling pathways involved in cell proliferation, death, migration, movement and vasculogenesis ($P < 0.05$, shared genes changed 1.25-fold in both siCon-siYAP1 and static-WSS comparisons; Supplementary Fig. 9a,b). Master regulators of these pathways included progesterone receptor (PGR), androgen receptor (AR), receptor tyrosine-protein kinase erbB-2 (*ERBB2*), hypoxia-inducible factor 1-alpha (*HIF1A*), platelet-derived growth factor-beta (*PDGFB*) and others. Components of the nuclear

factor kappa B (NFkB) signalling pathway were activated by WSS and inhibited by siYAP1, including transcription factor p65 (RELA) and TNF. Jun, Akt and Mek kinases were also implicated in regulation of genes contributing to cell movement.

*In silico* analysis identified enrichment of TEAD-binding sites in promoters of genes with established roles in cell migration, suggesting that YAP1 might be recruited to regulate their expression (Supplementary Fig. 10a). In fact, 34 of 36 genes linked to migration contained at least one TEAD1 (TEF-1) consensus site 1,500 base pairs upstream to 500 base pairs downstream of the transcriptional start site (TRANSFAC). TEAD4 was found to bind 27 of these genes by ChIP-seq (ENCODE Consortium), and 8 genes were previously identified as direct targets of YAP1 by ChIP-seq and gain or loss of function studies in cardiomyocytes and SF268 glioblastoma cells[21,22].

On the basis of this *in silico* analysis, YAP1 and TEAD4 can both be recruited to *TGFB2*, *TGM2*, *MYH9*, *MEF2C* and *ETS1*. A recent report has shown that over 80% of YAP1/TAZ/TEAD-binding to DNA occurs at enhancers located over 10 kb from the transcriptional start site of target genes in MDA-MB-231 breast cancer cells[23]. Of the 379 enhancer-regulated genes identified in that study, *ADRB2*, *ETS1*, *MYC*, *PLAUR* and *WWC1* from our list of 36 genes were shown by chromatin conformation assays and gene expression to be regulated by YAP1/TAZ/TEAD. Strikingly, 13 of the genes have been linked to cancer diagnosis and/or prognosis (ingenuity knowledge base) (Supplementary Fig. 10a). Twenty seven genes shared other invasive functions including chemotaxis, invasion, adhesion and formation of lamellipodia/filopodia (ingenuity knowledge base) (Supplementary Fig. 10b). In summary, our data indicate that fluid force present in the tumour microenvironment can trigger cancer cell motility and metastatic behaviour through modulation of gene expression downstream of ROCK–LIMK–cofilin-activated YAP1 signalling (Fig. 6). Further studies will be required to identify the direct effectors of YAP1 that are important for biomechanical induction of cell motility and understand how they prepare the cell for taxis.

## Discussion

Pathologic characteristics of prostate adenocarcinoma include changes in gland density, architecture of secretory ducts, presence of microvasculature, and composition and presumably volume and viscosity of prostatic fluid[46,47]. Many of these types of biophysical changes redefine genetic and paracrine signalling in the tumour microenvironment and the resident cancer cells. In early lymphatic metastasis, it has been suggested that passive intravasation can occur when primary tumour cells growing in a confined space push against each other, producing stress that can collapse lymphatic vessels and, potentially, force cells to breach fragile lymphatic vessels[48,49]. However, other factors such as cell

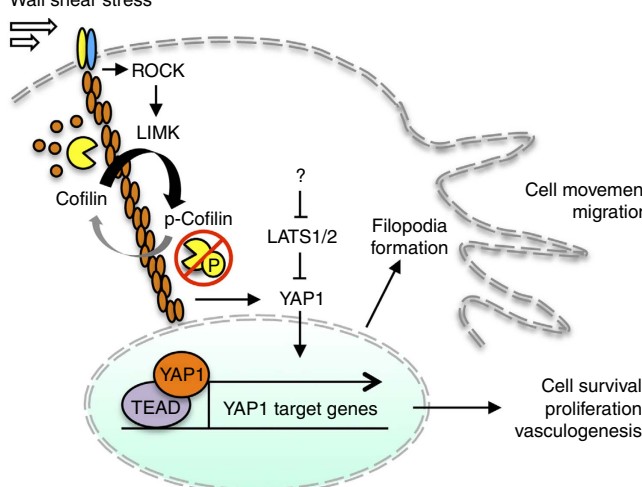

**Figure 6 | Model of signal transduction downstream of WSS.** Transduction of biomechanical force caused by fluid flow stimulates a signalling cascade through activation of ROCK and LIMK. Activated LIMK phosphorylates cofilin to inhibit its actin depolymerizing function. In the presence of stabilized F-actin, the cell is poised for regulation by LATS1/2. YAP1 is dephosphorylated and localizes to the nucleus to drive transactivation of gene promoters occupied by TEAD. Genes regulated downstream of YAP1 promote cell movement and survival. Although our data is consistent with LATS as a contributor to regulation, we cannot exclude alternate scenarios that include other kinases that post-transcriptionally modify YAP1.

chemotaxis and phenotypic adaptability point toward an active metastatic process. Within lymphatic vessels draining from the prostate, WSS caused by flow of lymph actively stimulates mechanosensors on cells lining the vascular lumen. WSS on the lymphatic endothelium drives lymphatic-directed immune cell trafficking by increasing CCL21 secretion and reorganizing PECAM1 and VE-Cadherin on the lymphatic endothelial cell surface[50,51]. Evidence suggests that this pathway could be utilized by cancer cells that overexpress chemokine cognate receptors[52]. In addition, heparan sulfate presented by lymphatic endothelial cells has been reported to play an important role as a mechanosensor in recruitment of cancer cells into peripheral lymphatic vessels, thus facilitating lymph node colonization via lymphatic metastasis[53]. Collectively, these reports suggest that lymphatic fluid flow provides cancer cells with biomechanical cues to navigate to the lymph nodes.

Herein, we model prostate cancer behaviour in the lymphatic vessels to identify factors modulated by mechanical cues. To mimic physiological ECM elasticity and fluid movement typical of reported shear stresses in the human lymphatics, we utilized custom-fabricated microfluidics generated by standard soft lithography techniques[54]. YAP1 is sensitive to soluble[55] and biophysical cues[6], such as elasticity of the extracellular matrix, cell shape and cell density. Thus, even routine cell culture conditions (that is, rigid tissue culture plastic) can artificially upregulate YAP1 and TAZ activity. The design of our biomimetic platform exposed cells to relatively low matrix stiffness ($\sim$1.7 MPa), which likely improved our ability to detect a YAP1 signature following application of low WSS. Further, evaluation of a range of WSS magnitudes enabled detection of enhanced motility responses not typical of cancer cell behaviour in the blood stream. We show that YAP1 and its downstream target genes were activated by 0.05 dyne cm$^{-2}$ WSS, corresponding to the velocity of fluid flow in the interstitium or initial lymphatics[56,57]. Importantly, this level of WSS regulates YAP1 activity to drive motility of prostate cancer cells.

As cell movement is required for tumour cell invasion into healthy tissue, our data suggest that WSS caused by flow could promote progression and metastasis of aggressive cancers. To transit forward during cell migration, actin polymerization must occur at the leading edge of the plasma membrane while actin filaments are disassembled at the rear. Here, we tested cofilin and gelsolin actin-severing proteins, and found that cofilin rapidly responds to WSS through dynamic changes in phosphorylation. Moreover, WSS generates more central F-actin compared with peripheral F-actin in PC3 cells (Supplementary Fig. 5a), consistent with stress fibre organization by ROCK[58]. In particular, when ROCK inhibitor was applied to evaluate upstream regulation of the cofilin pathway, re-phosphorylation of cofilin during WSS was interrupted, indicating that LIMK activity can be inhibited by Y27632. Consistent with these results, Wang *et al.*[30] showed that actin cytoskeleton and Rho GTPase are required for the decrease in S127 phosphorylation of YAP1 after release from energy stress. Taken together, our data support that ROCK triggers YAP1 activation and cell migration through regulation of the LIMK-cofilin pathway. It is known that inactivation of cofilin is necessary for motility[59], but cofilin is also essential for the formation of free barbed ends that are required for formation of lamellipodial protrusions by F-actin modulation[60,61]. In our study, cofilin knockdown significantly increased cellular motility. This partially reproduces results from a previous report showing that cofilin depletion enhanced cell migration and directionality[62]. Further, Aragona *et al.*[8] demonstrated that cofilin silencing significantly enhances YAP1/TAZ activity, supporting our observation that WSS-induced motility is mediated by YAP1.

Notably, we found that enhanced cell motility was driven by YAP1 and not TAZ, demonstrating that YAP1 and TAZ play distinct roles in response to biomechanical force. Although TAZ has 46% amino acid sequence identity with YAP1 and displays similar domain organization[63], TAZ lacks hydrophobic linker sequences between two α-helices in the amino-terminal TEAD-binding region, whereas two α-helices in YAP1, along with the connecting hydrophobic linker, are necessary for the growth promoting activity of the TEAD interaction[64,65]. Enhancement in cellular motility by YAP1 also requires this YAP1–TEAD interaction, supported by our data showing that YAP1 S127A/S94A did not significantly stimulate motility and the YAP–TEAD chemical inhibitor verteporfin abolished WSS-induced motility. As verteporfin has been shown to elicit YAP1-independent effects on tumour cell growth via STAT3 activation and autophagy[66], we further confirmed that the YAP1-selective peptide YTIP blocked motility. Although ANKRD1, CTGF and AMOTL2 have been reported to be downstream targets of YAP1/TAZ[23], these genes appear to be unrelated to motility in our study. Moreover, ANKRD1, CTGF and AMOTL2 were induced by WSS despite YAP1–TEAD disruption by verteporfin (Supplementary Fig. 7) and could be reduced only by silencing of TAZ (Supplementary Fig. 4a). These data reinforce that cell motility is determined by other downstream target(s) of YAP1. Here, we find that TEAD1-4 exhibit the ability to functionally compensate for the loss of individual TEAD members, thus depletion of all four was required to completely inhibit WSS response (Fig. 5e,f). Despite TEAD functional overlap, TEAD1 appeared to contribute uniquely to facilitating migration (Fig. 5f).

We identify YAP1-dependent gene expression changes induced by WSS. We found that 36 genes modified by YAP1 activity and WSS have executive roles in cell migration. Many of these genes have been implicated in regulation of hallmarks of cancer including invasiveness, migration and angiogenesis through modulation of MMPs, VE-cadherin, VEGF receptors and various kinase activities[67–69]. Most of the genes identified in our study also regulate chemotaxis, invasion, adhesion and formation of lamellipodia and/or filopodia in various types of cells and animals (Supplementary Fig. 10b). Taken together, our results suggest that a network of YAP1-dependent genes sensitive to WSS contribute to invasive cancer cell behaviour in the lymphatics.

In conclusion, we have demonstrated that the mechanical force generated by fluid flow regulates cellular behaviours fundamental to the process of metastasis. YAP1 was rapidly activated by low-level WSS typical of the lymphatic vasculature to drive cancer cell motility. Increased movement was dependent on YAP1–TEAD interactions downstream of the ROCK–LIMK–cofilin signalling axis. Our data support emerging evidence for the critical role that mechanical stimuli in the tumour microenvironment may play in cancer progression.

## Methods

**Cell culture and pharmacological reagents.** The PC3 human prostate cancer cell line was maintained in F-12K supplemented with 10% fetal bovine serum. DU145 and LNCaP human prostate cancer cells were maintained in RPMI 1640 supplemented with 10% fetal bovine serum at 37 °C in 5% $CO_2$ (v/v). HEK-293T cells were purchased from American Type Culture Collection (ATCC) and maintained in Dulbecco's modified essential medium (DMEM) supplemented with 10% fetal bovine serum. All culture media contained 1% penicillin and streptomycin antibiotics. Verteporfin (Sigma) was applied to cells at a concentration of 1 μM to disrupt the YAP1–TEAD interaction, Y27632 (Cayman Chemical) was used at 10 μM to selectively inhibit ROCK (Rho-associated coiled-coil containing protein kinase) and PF-562271 (Selleck Chemicals) was used at 5 μM with 2-h pre-incubation to inhibit FAK. LIMKi3 (Millipore) and U0126 (Cayman) were used at 10 μM. Cells were routinely passaged on tissue culture plastic and transferred into PDMS channels 24 h before application of WSS. Cell line authentication and mycoplasma negativity were confirmed by IDEXX.

**Microfluidic devices and application of WSS.** Microfluidic devices were prepared using standard soft lithography techniques[54]. Poly(dimethylsiloxane) (PDMS, Sylgard 184 silicone elastomer kit, mixed with a ratio of 10 wt% cross-linker; Dow Corning) was poured over a master mould, and incubated overnight at 60 °C. The cylindrical channels were 35 mm in length and 1.8 mm in diameter (Supplementary Fig. 1). Before seeding cells in the device, the channel was sterilized at 121 °C for 15 min and coated with 50 μg ml$^{-1}$ Type I Collagen at 4 °C for 24 h. Following a 24 h culture period on the PDMS surface, medium was injected into the channel and flushed at a constant flow rate $\gamma$ using a programmable syringe pump (PhD Ultra, Harvard Apparatus). Shear rate created at the inner surface of the channel was calculated as $\gamma = \frac{(m+2)f}{\pi r^3}$, and WSS ($\tau$) as, $\tau = \eta \gamma$, where $f$ is the total flow, $\gamma$ the internal radius of the channel and $\eta$ the fluid viscosity. The value of the dimensionless number $m$ depends on flow conditions. For laminar flow, $m = 2$ and for turbulent flow, $m > 2$ (ref. 70). We applied flow rates of 47 μl min$^{-1}$, corresponding to values of 0.05 dyne cm$^{-2}$ WSS. For immunostaining purposes, sticky-Slide VI$^{0.4}$ (ibidi GmbH, Germany) channels were attached to PDMS-layered coverslips coated with Type I Collagen before use.

**Ultrastructural analysis.** Cells were fixed with 2% glutaraldehyde for 1 h at room temperature and washed with deionized water for 5 min. Serial dehydration was done by stepwise incubation in 50, 75, 90, 95 and 100% for 10 min each. Cells were subsequently immersed in 1:1 ethanol/HMDS (hexamethyldisilazane, Sigma-Aldrich), 1:2 ethanol/HMDS and 100% HMDS for 5 min each. Images were obtained using a high-resolution field emission scanning electron microscope (Quanta 400 ESEM, FEI).

**Ectopic expression and knockdown.** The 8 × GTIIC-DsRed-Monomer reporter of YAP1/TAZ-TEAD activity was generated in two steps. First, the 8 × GTIIC-luc plasmid (Addgene) was sequenced, revealing that the 3′ UTR of luciferase was missing sequence upstream of the poly(A) hexanucleotide signal. The sequence was corrected to match the published sequence using a gBlock from IDT DNA. Next, the luciferase coding sequence was replaced with a PCR amplified fragment of DsRed-Monomer from pDsRed-Monomer-Hyg-N1 (Clontech) containing NcoI/FseI restriction sites. The final plasmid was sequence verified.

The pCMV-flag-YAP1 S127A, pCMV-flag-YAP1 5SA/S94A, pcDNA3-HA-TAZ and pcDNA3-HA-TAZ S89A constructs were obtained from Addgene, and transfection was performed using FuGENE6 (Promega). The pN1-eGFP-YTIP construct was kindly provided by W. Pu (Boston Children's Hospital) via W. J. Nelson (Stanford University)[43]. pGIPZ lentiviral shRNAs targeting Cofilin 1 (CFL1) and Gelsolin (GSN) were purchased from Thermo Scientific. All lentiviral supernatants were generated by transient transfection of HEK-293T cells with psPAX2 and pMD2.G as packaging and envelope plasmids, respectively, and collected 48 h after transfection. Supernatants were passed through a 0.45-μm filter and concentrated by Lenti-X concentrator (Clontech) to infect PC3 cells with the addition of 10 μg ml$^{-1}$ polybrene. SMARTpool siRNAs against YAP1, TAZ and TEAD1-4 were from Dharmacon. For siRNA transfection, cells were cultured in standard conditions and transfected using DharmaFECT 1 (Dharmacon). Briefly, cells were plated at 70% confluence and subjected to transfection the following day using 25 nM final concentration of each siRNA. The next day, cells were transferred to collagen-coated PDMS channels for WSS application.

**Time-lapse imaging.** For quantification of motility, PDMS channels or PDMS coated sticky channels were placed on an inverted microscope (Olympus IX-81) and cellular motility was observed with phase contrast microscopy under static or laminar WSS conditions. At a given flow rate, successive images were recorded every 3 min for 6 h in an environmental chamber maintained at 37 °C, 5% $CO_2$. Cells were tracked in time-lapse image sequences using the manual tracking plug-in for Image J 1.46r (http://rsb.info.nih.gov/ij). Image J output was integrated into Chemotaxis and Migration Tool software (ibidi GmbH, Germany) to determine migration position and velocity of cells in each channel. Migration data for each cell in one position were averaged over the entire cell population (>15 cells for each position, five positions per device and three or more devices for each condition).

For imaging of YAP1/TAZ-TEAD DNA binding activity, the 8 × GTIIC-DsRed-Monomer TEAD reporter was transfected into PC3 cells, and cells were imaged after 72 h under static or WSS conditions for 16 h in an environmental chamber. Both bright field and DsRed-Monomer images were acquired every 5 min using an exposure time for fluorescence of 500 ms. Images from each light channel were compiled into video files by Image J 1.46r. A rainbow RGB lookup table was applied to the video files using Image J.

**RNA extraction and quantitative RT-PCR.** Total RNA was isolated from channels using the RNeasy Micro kit (Qiagen), according to the manufacturer's instructions. Reverse transcription of RNA was performed using Applied Biosystems Multiscribe DNA polymerase, according to the manufacturer's instructions. Real-time Taqman PCR (Applied Biosystems) was performed in 10 μl reactions with primers provided by Applied Biosystems, according to the manufacturer's instructions. For calculation of fold change, cycle thresholds (Ct)

were determined using SDS 2.2.1 software (Applied Biosystems), and mRNA expression was normalized to GAPDH transcript and the control sample.

**Measurement of metalloprotease activity.** Following cell lysis, equal amounts of protein were mixed with assay buffer (50 mM Tris, 10 mM calcium chloride, 150 mM sodium chloride, 0.05% Brij-35, pH7.5) in a 100 μl reaction mixture. Total MMP fluorogenic peptide substrate (Mca-K-P-L-G-L-Dpa-A-R-NH$_2$, R&D system) was added to a 10 μM final concentration and incubated for 1 h at room temperature. The reaction mixture was read with 320 nm excitation and 405 nm emission using a fluorescence plate reader (Infinite M1000 Pro, TECAN).

**Gene expression profiling.** Total RNA was extracted with QIAGEN RNeasy kits for analysis of gene expression by Illumina Human HT-12 v4.0 Expression BeadChips. Data was checked for quality, background corrected and quantile normalized with GenomeStudio (Illumina). Analysis of differential expression was conducted with GenomeStudio for probe sets filtered by spot detection ($P < 0.01$) and difference in average signal intensity between treatment groups (dif > 20). Significantly changed genes were subjected to analysis of canonical pathways curated by Ingenuity (IPA, Ingenuity Systems). Hierarchical clustering and heatmap outputs from GenePattern (Broad Institute) included overlapping genes from differential expression analysis in GenomeStudio ($P$ value < 0.05) and direct downstream targets of YAP1 in the literature[21,22]. Gene regulatory networks were constructed with the IPA regulator effects algorithm ($P$ value < 0.05 cutoff, z-score > 2 cutoff) using gene sets filtered by significance (< 0.05) and fold-change (> 1.25) with nodes arranged by upstream regulators at top, target molecules in the middle, and disease or function at the bottom. Molecules experimentally determined to function in cell movement were compiled with BioProfiler (IPA), filtered by significance (< 0.05) and fold-change (> 1.25, and arranged by subcellular localization with Path Designer (IPA).

**TEAD consensus sequence and binding site analysis.** Prediction of TEAD consensus sequences was determined using the TRANSFAC knowledge-base (BIOBASE). The positional weight matrix for TEF-1 (TEAD1) was used with the MATCH algorithm to search DNA sequences for putative transcription factor binding sites 1,500 base pairs upstream and 500 base pairs downstream of the transcriptional start site within all promoters using similarity cutoffs to minimize the sum of false negatives and positives (core similarity cutoff of 0.693; matrix similarity cutoff of 0.887; false positive rate of 0.781).

ChIP-seq data for TEAD was compiled by interrogation of the hg19 assembly of the human genome (UCSC Genome Browser). Methylation and acetylation marks (H3K4Me3 and H3K27Ac) corresponding to promoter regions and active regulatory elements, respectively, were used in conjunction with the TEAD4 ChIP-seq track derived from ChIP-seq experiments performed by the ENCODE Project to identify experimentally defined TEAD-binding sites. All promoters were scored for TEAD4 peak numbers and signal intensity in H3K4Me3 and/or H3K27Ac regions within 20 kb of the transcriptional start site.

**Immunofluorescent staining of cultured cells.** Cells were fixed in 4% paraformaldehyde for 15 min and blocked by 5% bovine serum albumin in PBS-T (PBS with 0.1% Triton X-100) for 1 h at room temperature. Cells were treated with mouse anti-YAP1 monoclonal antibody (1:100 dilution, Abnova clone 2F12, Cat. No. H00010413-M01), anti-TAZ monoclonal antibody (1:100 dilution, BD pharmingen clone M2-616, Cat. No. 560235) and rabbit anti-tubulin polyclonal antibody (1:100 dilution, Santa Cruz clone H-300, Cat. No. sc-5546) diluted with 1% bovine serum albumin in PBS-T at 4 °C overnight, followed by Alexa 488-conjugated rabbit anti-mouse secondary antibody (1:500 dilution, Invitrogen, Cat. No. A11059), Cy3-conjugated donkey anti-mouse secondary antibody (1:500, Jackson Immunoresearch, Cat. No. 715-165-151) or Alexa 488-conjugated goat anti-rabbit secondary antibody (Cat. No. A11008). For F-actin, Alexa Fluor 594 Phalloidin (1:50 dilution, ThermoFisher, Cat. No. A12381) was used. Counter-staining for each condition was performed with Draq5 (Invitrogen). Images were captured by a Leica TCS SP5 confocal microscope with a Leica 63X oil objective lens (NA 1.4) and analysed with LAS Advanced Fluorescence software (Leica).

**Western blotting.** Cells were harvested in RIPA buffer (150 mM sodium chloride, 1% Triton X-100, 1% sodium deoxycholate, 0.1% SDS, 50 mM Tris–HCl, pH7.5 and 2 mM EDTA) with 1% protease and phosphatase inhibitor cocktails (Sigma). Equal amount of proteins were separated by SDS/PAGE and analysed by immunoblotting. Western blotting was prepared by standard procedures using anti-YAP1 (Santa Cruz clone H-9, Cat. No.sc-271134), phospho-YAP1 (Ser127; Cell Signaling Technology, Cat. No. 4911), anti-TAZ (BD Pharmingen clone M2-616), anti-cofilin (Abcam, Cat. No. ab42824), anti-cofilin (phospho-S3; Abcam, Cat. No. ab12866), anti-gelsolin (Santa Cruz ABS017, Cat. No. sc-57509), anti-phospho-p44/42 MAPK (phospho-ERK1/2, Thr202/Tyr204; Cell Signaling Technology, Cat. No. 9101), anti-p44/42 MAPK (ERK1/2; Cell Signaling Technology, Cat. No. 9102), anti-FAK (Cell Signaling Technology, Cat. No. 3285), anti-phospho-FAK (Tyr397; Cell Signaling Technology, Cat. No. 3283) and β-actin

(Santa Cruz clone C4, Cat. No.sc-271134) antibodies. The original scans of the blots are shown in Supplementary Fig. 11.

**Histopathology of primary tumours and metastases.** All animal experiments were performed according to the University of Texas Health Science Center Animal Welfare Committee guidelines for laboratory animals. PC3 cells stably expressing the DsRed-Express fluorescent protein gene reporter were implanted in mice to enable longitudinal imaging of metastasis from the prostate and 10–12 weeks post implant provided image guided resection and cancer assessment of excised axillary, renal, lumbar, inguinal, popliteal and sciatic lymph nodes, as described previously[27]. Briefly, 10$^6$ PC3-DsRed-Express cells were orthotopically implanted in the dorsal prostate of twenty 6–8-week-old male nu/nu mice (Jackson Labs). No formal randomization was employed for xenograft surgeries. Mice were fed a chlorophyll-free chow to avoid diet-induced autofluorescence from the red excitation light used for DsRed-Express fluorescent protein imaging. Of mice found to express DsRed-Express in the lymph nodes, tumour burden was verified histologically by H&E.

For histopathological scoring of YAP1 expression, slides were deparaffinized and treated in sodium citrate buffer pH 6 (Dako) for 10 min at 125 °C in a pressure cooker. Antibodies were incubated for 1 h against YAP1 (Acris, clone 1A12, 1:100, AM06727PU-N) at room temperature. MACH4 mouse probe and MACH4 universal polymer were applied for 15 min each, then followed by DAB (Dako) incubation. Counterstaining with CAT hematoxylin (Biocare Medical) was following by bluing reagent (StatLab Medical Products) for 15–30 s. Slides were rinsed with water, air-dried at 60 °C for 10–15 min, then coverslipped with EcoMount (Biocare Medical). YAP1 immunohistochemical score was determined by both per cent positivity and intensity of staining in PC3 tumour cells. Score was calculated as the product of per cent positive cells and staining intensity, with scores ranging from 0 to 300. As the YAP1 antibody was specific for human YAP1, mouse tissues did not show appreciable staining and were excluded from analysis. The pathologist was blinded to study design and hypothesis.

**Statistical analysis.** All data were analysed with SigmaPlot 12.5 for statistical significance and are reported as mean ± s.e.m. Differences in migration and protein localization were analysed with the two-tailed unpaired $t$-test. One-way ANOVA and the Holm–Sidak method for multiple comparisons were used to evaluate differences in motility and gene expression across three or more treatment groups. Where assumptions of normality were not met, the Kruskal–Wallis H and Dunn's post hoc tests were used. Significance levels of $P < 0.05$ and < 0.01 are denoted in graphs by a single asterisk * or double asterisks **, respectively. Representative results from at least three independent biological replicates are shown unless stated otherwise.

**Data availability.** Gene expression profile data have been deposited for public access in the NCBI Gene Expression Omnibus under Accession Number GSE73284. Other data and materials used for this study are available from the corresponding author on request.

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

## Acknowledgements

We thank S.-H. Lin and G. Gallick for critical discussions; G. Ayala for guidance in selection of YAP1 antibody and tissue staining protocols; and H. Robinson for preparation of the orthotopic animal model. The YTIP plasmid was a kind gift from W.J. Nelson. This work was funded by grants from the State of Texas Emerging Technology Fund, American Society of Hematology Scholar Award, Mission Connect: a Program of the TIRR Foundation, and National Institutes of Health (K01DK092365) to P.L.W. Flow cytometry instrumentation was supported in part through the Cancer Prevention and Research Institute of Texas (RP110776).

## Author contributions

H.J.L. and P.L.W. designed the study, analysed the data, wrote the manuscript and directed the research. H.J.L, M.F.D., K.M.P. and J.A.O performed experiments. E.M.S.-M. directed the animal study and provided primary tumour and lymph node samples. S.Z. conducted all pathological scoring and J.P.H. developed the TEAD reporter system.

## Additional information

**Competing financial interests:** The authors declare no competing financial interests.

