## [Peer Review File · Nature Communications]

Reviewer #1 (Remarks to the Author)

This manuscript identifies a contribution of Yap to the influence of shear stress on cell motility. Using microfluidic chambers, the authors found that Yap is activated in PC3 cells (a prostate cancer cell line) in response to wall shear stress (WSS). This activation was suppressed by a ROCK inhibitor, and associated with regulation of cofilin. They went on to show that Yap activation promotes cell motility, and identified genes whose expression is induced by Yap in cells subject to WSS. Yap has previously been implicated in metastatic behavior, and ROCK and cofilin have been shown in previous studies to influence Yap. The novelty of this study, then, is in linking these together to show that WSS affects Yap, and thereby cell motility. For the most part the results are clear and documented appropriately, but I do have a few concerns:

1) The authors claim that the influence of WSS depends upon Yap, but not on Taz. The data included to support this is not convincing. In vivo studies in mice have shown that Yap and Taz are partially redundant. The authors show that a knockdown of Yap reduced cell movement, whereas a knockdown of Taz did not, but this could simply reflect the sensitivity of cell migration to the total levels of Yap/Taz, together with differences in expression levels. So they need to be more cautious in their conclusions here, or come up with more compelling experiments.

2) The last section, identifying genes regulated by WSS and Yap, is weak because it's purely descriptive. We are given a set of genes that are regulated, but no evidence that these particular genes contribute to the migration phenotype observed.

3) It would add to the manuscript if the authors could provide more insight into Yap regulation in their system. For example, Yap is regulated by cell density, cell size, and cell shape. Could the influence of WSS on Yap simply be a consequence of changes in cell shape (and consequent effects on the cytoskeleton)? Does the influence of WSS depend on Integrin-ECM contacts? Their study identifies some kind of effect on the cytoskeleton, but I would find it more interesting (as the cytoskeleton is well known to regulate Yap) if they could clarify whether the connection between WSS and Yap involves previously described inputs, or is there something new here?

Reviewer #2 (Remarks to the Author)

This is an interesting MS where authors have modeled the effect of vascular hemodynamic stress on cancer cells using engineered platforms. These multidisciplinary efforts are critical if we aim to truly model in vitro the real complexity of cancer biology and metastatic dissemination. I therefore think that the MS is interesting but nevertheless does require some revision.

Above all, the key experiments must be replicated with independent shRNAs or siRNAs for all the main players at stake. Moreover, there are various other issues to be addressed:

1) I am not sure that they are really looking at Hippo signaling here. The statement: " Notably, components of the HIPPO signaling pathway were differentially expressed, including several phosphatases (PPP1R3C, PPP2CB, PPP2CA, and PPP2R3B), transcription regulators (WWC1 and SMAD3), and signal transduction factors (YWHAB, YWHAE, and YWHAG)" ...simply does not make any sense, and should be deleted.

2) Fig. 2c. I cannot appreciate any TAZ stabilization and a minor effect on P-YAP. overall, not really convincing. Is this sufficient to explain what they observe?

3) More generally, I understand that there are changes in YAP S127 phosphorylation but they should not be overinterpreted. Functional data are not consistent with a role of LATS in the cofilin regulation that they are investigating. This is further complicated by what shown in Fig 6B: Lats mRNA is Lower in High YAP: this may (or not) be sufficient to explain why in YAP ON cells there is also low P-YAP, but indicating a transcriptional regulation (or another regulation at the Lats transcript stability level) that, if significant, would be obviously unrelated to any Hippo signaling

(involving the tuning of LATS kinase activity and not of LATS expression).

4) When they state "155 Examination of genes found previously by ChIP-Seq to be bound by YAP1 and those classically defined as YAP1/TAZ targets revealed".....Please add References. And explain how the experiment (wet or in silico) was done.

5) On verteporfin: 258 Application of a small molecule..... This drug is great for protein-protein interactions in vitro, but on live cells it can be non-specifically toxic and its effects have been recently shown in Science Signaling to be YAP-independent. This needs to be deleted, and replicated with better reagents.

6) Figure: On "Cofilin, and gelsolin were detected with nuclei by immunofluorescence in PC3 cells." This is not convincing and not really useful; it is quite secondary for this MS to know where are these acting in the cell. Please delete.

7) Movies 3-4 are hard to appreciate, the quality is very low. They should try to improve the visualization of YAP activity to appreciate how this physical stimulation affects a transcription factor in live cells over time.

8) 2F-I, 3c, e, g; and figure 5: The technicality of the experiment and its significance need to be detailed in the text more clearly.

9) Fig 4d-e: limk-siRNA has no effect, and there may be many reasons for it. But at face value this is confounding. And note that this is not inconsistent with the effect of the Rock inhibitor, as Rock can also signal through MLCK, and not only LIMK. In sum, Please delete. They may call this "ROCK-Cofilin" pathway

10) Fig 4c: the effect of Cofilin-siRNA cannot be referred to S. Piccolo's work, Aragona et al. The authors have to prove that in their own hands and present context that YAP is downstream of Cofilin inactivation, by showing a rescue after using double/combined siRNAs Cofilin + siRNA YAP.

11) Figures 6c and d are really stretching too thin. Without any validation (and this should be extensive, not just the classic targets), these represent the products of a pure bioinformatic exercise involving merging of publicly available TEAD ChIPSeq, with publicly available YAP ChIP-Seq (most likely done in cell types that are not those here studied) and a further bioinformatic prediction of the promoters regulated by these putative YAP-chromatin-binding sites... (see Methods: there is no real experiment here just cross-matching of data in silico). In sum, a train of unvalidated assumptions, wishful-thinking etc. They should also know that many YAP/TAZ target genes are regulated by distant superenhancers, as shown by the Camargo, Piccolo and Bauer groups. There is no evidence that those genes are real target or real direct targets. In any case this should go for an online supplementary figure.

12) Discussion is a bit too long. The discussion on the various TEADs, on their role in embryos can be deleted. I also found the whole paragraph on ETS and TGFb, in absence of any functional involvement, really far fetched and the same is true for PLAUR. Please delete.

Reviewer #3 (Remarks to the Author)

A. In this manuscript, the authors identify a role for YAP1 in promoting cancer cell motility at shear stress levels representative of flow in lymphatic vessels, but not blood vessels. To conduct these experiments, the authors used a microfluidics platform of biologically relevant stiffness (~1 MPa). Overall, the authors have performed a thorough and comprehensive study supporting their hypothesis that YAP1, but not TAZ, is important to shear stress-mediated cancer cell motility. This response is mediated by ROCK, LIMK and YAP1-TEAD interactions.

B. The authors are addressing an important but not well-studied aspect of cancer cell metastasis. Metastasis via the lymphatics is an important process in many cancers, but there are numerous unanswered questions about the biology of this process. The authors shed some light on one aspect of this process.

C. I particularly like the use of a softer substrate to better mimic vascular biomechanics and account for YAP1's sensitivity to substrate stiffness; not accounting for the non-physiological effects of very stiff substrates (e.g. plastic, glass) is a deficit of many in vitro studies.

D. One overarching question I have is regarding the statistical significance of the data presented

versus the effect sizes. With the sample sizes being used, relatively small differences can be statistically significant, but I am uncertain in some of the presented cases whether the magnitude of the effect is biologically important. As an example, I am not sure the data showing cofilin's involvement is especially compelling (in contrast, the LIMK and TEAD data are much more clear cut). It would be valuable if the authors addressed this point.

E. Another general question I have is to what extent active cell motility, versus passive transport (by fluid flow) is important in the movement of cancer cells through lymphatics. Is this known? It would be useful to discuss whatever is known about this.

F. I have several specific comments to address / consider:

a. In Figure 2G and 2I, it would be good to include static data for the siYAP1, siTAZ, siAMOTL2, and siCTGF conditions.

b. In Figure 4A, there appears to be an increase in cofilin intensity with WSS; however, this is not in evidence in the western blot data. Did the authors observe any increase in cofilin levels with WSS?

c. In Figure 4C, when cofilin or gelsolin are silenced, is there still a significant increase in motility with WSS? If not, do the authors have an explanation for this change from the control condition?

d. In Figure 4G, what is the effect of Y27632 on static motility? Is the effect of the ROCK inhibitor to decrease motility under all conditions, or only under WSS stimulation?

e. From my perspective, Figure 6 seems to come a little bit from left field. I understand the authors' desire to include it, but it does not seem to fit particularly well the rest of the nice mechanistic data the authors have presented. I think the manuscript would be just as strong if the current Figure 6 was omitted (really up to the authors).

G. The appropriate work is cited.

H. The manuscript is clear and well-written.

Reviewers' comments:

Reviewer #1 (Remarks to the Author):

This manuscript identifies a contribution of Yap to the influence of shear stress on cell motility. Using microfluidic chambers, the authors found that Yap is activated in PC3 cells (a prostate cancer cell line) in response to wall shear stress (WSS). This activation was suppressed by a ROCK inhibitor, and associated with regulation of cofilin. They went on to show that Yap activation promotes cell motility, and identified genes whose expression is induced by Yap in cells subject to WSS. Yap has previously been implicated in metastatic behavior, and ROCK and cofilin have been shown in previous studies to influence Yap. The novelty of this study, then, is in linking these together to show that WSS affects Yap, and thereby cell motility. For the most part the results are clear and documented appropriately, but I do have a few concerns:

1) The authors claim that the influence of WSS depends upon Yap, but not on Taz. The data included to support this is not convincing. In vivo studies in mice have shown that Yap and Taz are partially redundant. The authors show that a knockdown of Yap reduced cell movement, whereas a knockdown of Taz did not, but this could simply reflect the sensitivity of cell migration to the total levels of Yap/Taz, together with differences in expression levels. So they need to be more cautious in their conclusions here, or come up with more compelling experiments.

Comment is addressed in main body of text lines 217-218; data is added in Extended Data Figure 3a.

To clarify the role of TAZ in WSS-induced motility and to supplement any YAP/TAZ deficiency as a result of YAP knockdown, wild type TAZ and a constitutively active form of TAZ (TAZ S89A) were overexpressed in siYAP knockdown cells. Cellular motility was evaluated and was found not to differ from siYAP alone. This supports the notion that TAZ does not functionally compensate for YAP in the context of WSS-induced motility. This result was added in Extended Data Figure 3, showing that neither wild

type TAZ nor constitutively active TAZ affects cellular motility.

2) The last section, identifying genes regulated by WSS and Yap, is weak because it's purely descriptive. We are given a set of genes that are regulated, but no evidence that these particular genes contribute to the migration phenotype observed.

Comment has been addressed.

Ingenuity based analyses were moved from Figure 6 to Extended Data Figures 7 and 8, consistent with recommendation by reviewer 2.

3) It would add to the manuscript if the authors could provide more insight into Yap regulation in their system. For example, Yap is regulated by cell density, cell size, and cell shape. Could the influence of WSS on Yap simply be a consequence of changes in cell shape (and consequent effects on the cytoskeleton)? Does the influence of WSS depend on Integrin-ECM contacts? Their study identifies some kind of effect on the cytoskeleton, but I would find it more interesting (as the cytoskeleton is well known to regulate Yap) if they could clarify whether the connection between WSS and Yap involves previously described inputs, or is there something new here?

Comments addressed in text lines 227-228 and 261-263, and associated data is added in Figure 4h, 4i, and Extended Data Figure 4a.

Our analyses show enrichment in central F-actin by WSS after 3 hr. Two previous reports show that Rho kinase regulates central F-actin (references added). Additionally, we have added data showing that LATS1 activity is decreased by WSS through dephosphorylation, whereas inhibition of Rho kinase increases LATS1 activity in the presence of WSS. Together, these data suggest that WSS induces cytoskeletal rearrangement through the Rho kinase-LIMK-Cofilin signaling axis, followed by LATS1 inactivation and YAP activation. Although we found that LATS1 activation can be regulated by Rho kinase, we need to investigate further how F-actin rearrangement affects LATS1 activity in future studies.

Reviewer #2 (Remarks to the Author):

This is an interesting MS where authors have modeled the effect of vascular hemodynamic stress on cancer cells using engineered platforms. These multidisciplinary efforts are critical if we aim to truly model in vitro the real complexity of cancer biology and metastatic dissemination. I therefore think that the MS is interesting but nevertheless does require some revision.

Above all, the key experiments must be replicated with independent shRNAs or siRNAs for all the main players at stake. Moreover, there are various other issues to be addressed:

1) I am not sure that they are really looking at Hippo signaling here. The statement:

" Notably, components of the HIPPO signaling pathway were differentially expressed, including several phosphatases (PPP1R3C, PPP2CB, PPP2CA, and PPP2R3B), transcription regulators (WWC1 and SMAD3), and signal transduction factors (YWHAB, YWHAE, and YWHAG)" ...simply does not make any sense, and should be deleted.

Comment is addressed in text lines 156-159: sentence reworded to soften the claim that HIPPO signaling is altered.

2) Fig. 2c. I cannot appreciate any TAZ stabilization and a minor effect on P-YAP. overall, not really convincing. Is this sufficient to explain what they observe?

As Ser-127 is one of many phosphorylation sites that regulate YAP activity, it may be limited in its ability to inform upon functional consequences of WSS. In the absence of Western blot data, immunofluorescent (Figure 3a,3b) and functional analyses (siRNA, YTIP, verteporfin, and YAP-S127A overexpression) strongly implicate YAP in driving WSS-induced motility by a mechanism that involves nuclear localization of YAP and subsequent transactivation of YAP target genes that modify cell movement.

As reviewer 2 has suggested, Western blot data shows a modest increase in TAZ protein by WSS. However, we also show that its downstream target genes such as AMOTL2, ANKRD1 and CTGF are dramatically reduced by TAZ siRNA (Extended Data Figure 3b) and significantly increased by WSS (Figure

2c; Extended Data Figure 3b). By immunofluorescence, we show that TAZ localized to the nucleus in response to WSS (Figure 3a, 3b). Overall, TAZ was functionally activated by WSS.

3) More generally, I understand that there are changes in YAP S127 phosphorylation but they should not be overinterpreted. Functional data are not consistent with a role of LATS in the cofilin regulation that they are investigating. This is further complicated by what shown in Fig 6B: Lats mRNA is Lower in High YAP: this may (or not) be sufficient to explain why in YAP ON cells there is also low P-YAP, but indicating a transcriptional regulation (or another regulation at the Lats transcript stability level) that, if significant, would be obviously unrelated to any Hippo signaling (involving the tuning of LATS kinase activity and not of LATS expression).

Comments are addressed in text lines 261-263

As reviewer 2 indicated, LATS transcript level showed a non-significant reduction with WSS (high YAP). Instead of regulation of mRNA transcript levels, we found that LATS activity is decreased by WSS via Western blot analysis of post-translational modification (phospho-Thr 1079). This new data was added in Figure 4h, and is consistent with low LATS activity and decreased YAP phosphorylation following WSS. Further, when ROCK inhibitor is present, LATS activity was restored, corresponding with increased phospho-S127 YAP levels (Figure 4f, 4i). This data supports that ROCK-Cofilin-LATS signaling is critical to regulation of YAP activity and subsequent motility downstream of WSS.

4) When they state "155 Examination of genes found previously by CHIP-Seq to be bound by YAP1 and those classically defined as YAP1/TAZ targets revealed".....Please add References. And explain how the experiment (wet or in silico) was done.

Manuscript text lines 156-162 were edited.

We include explanation that the analyses we conducted were in silico, based upon the Ingenuity Knowledge Base (now referenced).

5) On verteporfin: 258 Application of a small molecule..... This drug is great for protein-protein interactions in vitro, but on live cells it can be non-

specifically toxic and its effects have been recently shown in Science Signaling to be YAP-independent. This needs to be deleted, and replicated with better reagents.

The YAP-TEAD inhibitory peptide (YTIP) has been tested and data added in Extended Data Figure 5c. Results are consistent with siYAP knockdown and verteporfin treatment.

Description has been added to text lines 276-279 and in methods section.

Based upon discussion with Dr. James Nelson, the corresponding author of Benham-Pyle et al., Science 348:1024 (2015) describing involvement of YAP in mechanically induced cell cycle entry, there may be interest from other reviewers and readers in cellular response to verteporfin. If deemed necessary by the reviewers and editor, those data can be removed upon request.

6) Figure: On "Cofilin, and gelsolin were detected with nuclei by immunofluorescence in PC3 cells." This is not convincing and not really useful; it is quite secondary for this MS to know where are these acting in the cell. Please delete.

Data has been deleted.

7) Movies 3-4 are hard to appreciate, the quality is very low. They should try to improve the visualization of YAP activity to appreciate how this physical stimulation affects a transcription factor in live cells over time.

Supplementary Movies 3 and 4 have been replaced.

8) 2F-l, 3c, e, g; and figure 5: The technicality of the experiment and its significance need to be detailed in the text more clearly.

Description and interpretation of data has been added in text 122-124, 255-259, 293-297, and in the methods section.

9) Fig 4d-e: limk-siRNA has no effect, and there may be many reasons for it. But at face value this is confounding. And note that this is not inconsistent

with the effect of the Rock inhibitor, as Rock can also signal through MLCK, and not only LIMK. In sum, Please delete. They may call this "ROCK-Cofilin" pathway

The model of our proposed signaling cascade is depicted for greater clarity in Figure 6c. Briefly, LIMK is an actin-binding kinase that phosphorylates cofilin to inactivate its actin-severing function. We show that treatment with a compound-based LIMK inhibitor (LIMKi3) reduces phospho-S3 cofilin (Figure 4d). Mechanistically, we expect that LIMKi3 allows unchecked activation of cofilin, depolymerization of F-actin, and subsequent elevated levels of inactive phospho-S127 YAP. This is consistent with increased phospho-YAP seen in WSS with LIMKi3 (Figure 4d) and the failure of WSS to stimulate motility in the presence of LIMKi3 (Figure 4e).

10) Fig 4c: the effect of Cofilin-siRNA cannot be referred to S. Piccolo's work, Aragona et al. The authors have to prove that in their own hands and present context that YAP is downstream of Cofilin inactivation, by showing a rescue after using double/combined siRNAs Cofilin + siRNA YAP.

Comment addressed in text lines 235-238; new data has been added in Figure 4c

Knockdown of YAP reduced motility that results from cofilin knockdown. This result demonstrates a rescue, and specifically that YAP lies downstream of cofilin to modulate taxis.

11) Figures 6c and d are really stretching too thin. Without any validation (and this should be extensive, not just the classic targets), these represent the products of a pure bioinformatic exercise involving merging of publicly available TEAD ChIPSeq, with publicly available YAP ChIP-Seq (most likely done in cell types that are not those here studied) and a further bioinformatic prediction of the promoters regulated by these putative YAP-chromatin-binding sites... (see Methods: there is no real experiment here just cross-matching of data in silico). In sum, a train of unvalidated assumptions, wishful-thinking etc. They should also know that many YAP/TAZ target genes are regulated by distant superenhancers, as shown by the Camargo, Piccolo and Bauer groups. There is no evidence that those genes are real target or real direct targets. In any case this should go for an

online supplementary figure.

In silico data has been moved to Extended Data Figures 7 and 8.

12) Discussion is a bit too long. The discussion on the various TEADs, on their role in embryos can be deleted. I also found the whole paragraph on ETS and TGFb, in absence of any functional involvement, really far fetched and the same is true for PLAUR. Please delete.

These sections have been deleted.

Reviewer #3 (Remarks to the Author):

A. In this manuscript, the authors identify a role for YAP1 in promoting cancer cell motility at shear stress levels representative of flow in lymphatic vessels, but not blood vessels. To conduct these experiments, the authors used a microfluidics platform of biologically relevant stiffness (~1 MPa). Overall, the authors have performed a thorough and comprehensive study supporting their hypothesis that YAP1, but not TAZ, is important to shear stress-mediated cancer cell motility. This response is mediated by ROCK, LIMK and YAP1-TEAD interactions.

B. The authors are addressing an important but not well-studied aspect of cancer cell metastasis. Metastasis via the lymphatics is an important process in many cancers, but there are numerous unanswered questions about the biology of this process. The authors shed some light on one aspect of this process.

C. I particularly like the use of a softer substrate to better mimic vascular biomechanics and account for YAP1's sensitivity to substrate stiffness; not accounting for the non-physiological effects of very stiff substrates (e.g. plastic, glass) is a deficit of many in vitro studies.

D. One overarching question I have is regarding the statistical significance of the data presented versus the effect sizes. With the sample sizes being used, relatively small differences can be statistically significant, but I am uncertain in some of the presented cases whether the magnitude of the

effect is biologically important. As an example, I am not sure the data showing cofilin's involvement is especially compelling (in contrast, the LIMK and TEAD data are much more clear cut). It would be valuable if the authors addressed this point.

Data has been added in Figure 4c; comments in text lines 235-238

Biological relevance of the changes induced in YAP activity by WSS is an important and challenging aspect of our studies. In the current manuscript, we document only one function of YAP in regulation of motility. Based upon observations from network analysis (Extended Data Figure 7), we conclude that YAP also mediates effects of WSS on apoptosis, cell growth, and proliferation. These other biological functions, in concert with altered taxis, may have important implications for resilience of tumor cells during metastasis, as supported by our observation that a greater number of metastases in the lymph nodes express higher levels of YAP (Figure 2e, 2f). This potential biological relevance is detailed in the discussion section.

To clarify specifically the involvement of cofilin in response to WSS, we employed double knockdown of cofilin and YAP (new data in Figure 4c). As shown in Figure 4b and 4c, cofilin knockdown enhances cell migration equally under static and WSS conditions. Enhanced motility by cofilin knockdown was significantly reduced by siYAP (to velocity typical of static cultures), further implicating YAP downstream of cofilin.

E. Another general question I have is to what extent active cell motility, versus passive transport (by fluid flow) is important in the movement of cancer cells through lymphatics. Is this known? It would be useful to discuss whatever is known about this.

Brief discussion and references are provided to describe an alternate hypothesis that has been proposed for passive intravasation of tumor cells into the lymphatics in text lines 349-354.

F. I have several specific comments to address / consider:

a. In Figure 2G and 2I, it would be good to include static data for the siYAP1, siTAZ, siAMOTL2, and siCTGF conditions.

Static results have been added in Figure 3g, 3h, and 3i.

b. In Figure 4A, there appears to be an increase in cofilin intensity with WSS; however, this is not in evidence in the western blot data. Did the authors observe any increase in cofilin levels with WSS?

The phosphorylation level of cofilin was transiently increased by WSS (peak at 5 min; see Figure 4f), but total cofilin protein levels were not changed. We have removed the immunofluorescent images that appeared to show elevated cofilin abundance, according to recommendation by reviewer 2.

c. In Figure 4C, when cofilin or gelsolin are silenced, is there still a significant increase in motility with WSS? If not, do the authors have an explanation for this change from the control condition?

We find that enhanced velocity caused by cofilin knockdown in static conditions is not further increased when cells are exposed to WSS (static and WSS are statistically equivalent in CFL1 shRNA cells in Figure 4b and c). This is consistent with the idea that WSS acts through cofilin to drive cell movement.

d. In Figure 4G, what is the effect of Y27632 on static motility? Is the effect of the ROCK inhibitor to decrease motility under all conditions, or only under WSS stimulation?

Static motility has been added to Figure 4g and demonstrates equivalent velocity of Y27632-treated cells in static and WSS conditions.

e. From my perspective, Figure 6 seems to come a little bit from left field. I understand the authors' desire to include it, but it does not seem to fit particularly well the rest of the nice mechanistic data the authors have presented. I think the manuscript would be just as strong if the current Figure 6 was omitted (really up to the authors).

Data have been moved to Extended Data Figures 7 and 8.

G. The appropriate work is cited.

H. The manuscript is clear and well-written.

Reviewer #1 (Remarks to the Author)

I'm satisfied with the authors revisions.

Reviewer #2 (Remarks to the Author)

REV2.

The MS unfortunately did not advance when compared to the previous version. I fear that they are not taking real advantage of the referees' advices. Most of the time, Revs asked to rephrase, to remove, to unload (and this was not done), and not to add more incomplete set of data and stronger, not entirely validated conclusions (as they did). I still support this MS but I would like to see a revised MS containing a point-by-point response to what listed below (here and there disappointingly restated from my previous report).

1) YAP vs TAZ. Conclusions are not compelling and, in any case, statements too strong. Negative data are simply negative data. For example: " YAP1 knockdown did 220 not reduce AMOTL2 or CTGF transcripts (Extended Data Fig. 3b) and their knockdown failed to reduce cellular velocity (Fig. 3h, i). Thus, these YAP1/TAZ target genes appear to be regulated chiefly by TAZ. This conclusion is flawed as there may be YAP or TAZ independent ways to control those genes. By no mean they are YAP/TAZ specific markers.

Moreover, this contrasts with the validation of TAZ in WSS See response to Ref2, point2.

Advice: there is no need to stress the difference. The negative data on TAZ can be deleted. TAZ remains to be investigated.

2) The authors are not doing the few, key experiments requested. I am referring to the YAP target genes (and lines 156-164). (point 2 ref 1 and points 1 and 4 of ref2). A) Ingenuity Knowledge Base can be at the most used to start, as a suggestion. But not taken as a fact. B) Validation is mandatory. As such, on my former point 4

"Examination of genes found previously by ChIP- Seq to be bound by YAP1 and those classically defined as YAP1/TAZ targets revealed"

this remains a flawed statement. The authors should validate these genes by Chip-PCR in their own cells and conditions. Alternatively, it would be easier to stay with obvious YAP/TAZ targets CTGF-CYR-ANKRD1 (and deleted the others) and on one or two of them show the differential recruitment of YAP by Chip-PCR after WSS. I have no idea what Dicer means, for example (see also Fig6, where actual checking of Dicer levels by qRTPCR (same in WSS with and without YAP) do not validate the microarray (6a)). And neither Dicer nor others of the genes listed in 6a fits the notion of established YAP/TAZ targets.

In other words: I do appreciate the YAP hype on cell mechanics and Hippo signaling and I am fully in favor of data driven hypotheses if they really to end up on YAP. But they are really stretching this approach to the limit; it seems to me "data driven with a bias" (= canalizing results toward the obvious YAP culprit). For example, it would be nice, if possible, to offer a revalidation by searching an overlap between bona fide YAP targets taken from the lists of direct targets that I saw recently published by a few groups (including Novartis and F. Camargo's labs).

Similarly, the phrase copied on my former point 1: "Notably, components of the HIPPO signaling pathway were differentially expressed, including several phosphatases (PPP1R3C, PPP2CB, PPP2CA, and PPP2R3B), transcription regulators (WWC1 and SMAD3), and signal transduction factors (YWHAB, YWHAE, and YWHAG)"

(and the title of their paragraph, see 139)

continues to make absolutely no sense to me. This can only be deleted, in absence of experimental validations.

3) Rev.1 asked about potential integrin connection (her/his point 3) and see my own Rev2 point 3. I was not convinced about the Y127 phosphorylation (a LATS target residue) or LATS data, and I am even less convinced now. I had not asked for any experiment. I just suggested to attenuate

and be more careful in the conclusions. Phospho-YAP as read-out is fine, but it is a downstream step of a series of events that they do not know (and do not need to address here). In contrast in this revision they add a western on LATS1 phosphorylation that complicates the message into an unexplored area, simply shifting the problem to an upper level (with its own issues) without really solving much.

A) the Ab is not validated by siRNAs against LATS1. Blots are not normalized to total LATS1. And LATS2?

B) Even if data of point A were present, the data would remain purely correlative, without making a pathway, as the authors are instead suggesting in the diagram of Fig.6.

C) bringing UP LATS phosphorylation means that we need to bring up regulation on its own kinase, that is (mainly) MST etc. All untested steps and regulations. Something for a different and dedicated paper.

D) the authors do not know the function of LATS1, the role of LATS2, nor the functional role of YAP phosphorylation. There is no causality between these inferred biochemical interactions and the biology here investigated. Certainly there are no connections with cofilin etc. There is no proof that LATS mediate the effects of Rock-cofilin.

E) Although the rescue of LATS-phosphorylation by Rock inhibitor is interesting, the data does not discriminate between a direct regulation, a parallel event or a feedback. As such, the diagram of Figure 6 remains flawed. F) I suggest to delete the new data on LATS1 and to focus on the regulation of ROCK by integrin or other contextual mechanical signals (Shape, rigidity etc).

In sum: this remains a potentially interesting paper for Nat. Comm. However, they should offer a new revision in which they focus and word carefully their conclusions, remove unnecessary and confounding data, and provide clear validation of their data-driven entry point.

Reviewer #3 (Remarks to the Author)

In this manuscript, the authors identify a role for YAP1 in promoting cancer cell motility at shear stress levels representative of flow in lymphatic vessels, but not blood vessels. To conduct these experiments, the authors used a microfluidics platform of biologically relevant stiffness (~1 MPa). Overall, the authors have performed a thorough and comprehensive study supporting their hypothesis that YAP1, but not TAZ, is important to shear stress-mediated cancer cell motility. This response is mediated by ROCK, LIMK and YAP1-TEAD interactions. The authors are addressing an important but not well-studied aspect of cancer cell metastasis. Metastasis via the lymphatics is an important process in many cancers, but there are numerous unanswered questions about the biology of this process. The authors shed some light on one aspect of this process. In the revised manuscript, the authors have included additional control data that address my concerns and have strengthened the manuscript.

We greatly appreciate the time that all reviewers have invested in providing us with constructive feedback. We better recognize the importance of distinguishing between YAP1-responsive genes and bona fide transcriptional targets. We are now acquainted with some of the recent YAP1 and Hippo studies that begin to set new standards for analysis of transcription factors and cofactors. Along with this new appreciation, we also recognize the limitations of our current microfluidics system in terms of its ability to provide sufficient cellular material for these types of analyses. Therefore, in addition to new experimental data and new computational analysis, we reword text throughout the manuscript to soften claims about unvalidated and unconfirmed YAP1 target genes. Below we outline specific changes that have been made. We additionally provide a version of the text with **changes in blue bold text placed at the END** of the submission materials.

Reviewers' comments:

Reviewer #1 (Remarks to the Author):

I'm satisfied with the authors revisions.

Reviewer #2 (Remarks to the Author):

REV2.

The MS unfortunately did not advance when compared to the previous version. I fear that they are not taking real advantage of the referees' advices. Most of the time, Revs asked to rephrase, to remove, to unload (and this was not done), and not to add more incomplete set of data and stronger, not entirely validated conclusions (as they did). I still support this MS but I would like to see a revised MS containing a point-by-point response to what listed below (here and there disappointingly restated from my previous report).

1) YAP vs TAZ. Conclusions are not compelling and, in any case, statements too strong. Negative data are simply negative data. For example: " YAP1

knockdown did not reduce AMOTL2 or CTGF transcripts (Extended Data Fig. 3b) and their knockdown failed to reduce cellular velocity (Fig. 3h, i). Thus, these YAP1/TAZ target genes appear to be regulated chiefly by TAZ. This conclusion is flawed as there may be YAP or TAZ independent ways to control those genes. By no means they are YAP/TAZ specific markers.

We have retracted claims that these genes are specifically TAZ target genes and have restructured the paragraph on these data for greater clarity. Following motility plots for siYAP and siTAZ, we reference the data showing that siTAZ significantly reduces ANKRD1 and CTGF transcript levels. Rather than focus on the difference between YAP1 and TAZ, we present this data to support that TAZ is activated by WSS but is silenced by siTAZ, consistent with Western blots showing that knockdown of TAZ is effective (Fig. 3e). “Despite the failure of TAZ siRNA to interrupt motility, our data was consistent with TAZ activation by WSS, as knockdown of TAZ suppressed WSS-induced increases in ANKRD1 and CTGF (Supplementary Fig. 4a). YAP1 knockdown did not reduce WSS-induced ANKRD1 or CTGF transcript levels.”

To clarify, the data referenced in Supplementary Figure 4b shows that siTAZ but not siYAP1 (Dharmacon SMARTpool) significantly reduces ANKRD1 and CTGF transcript levels.

Moreover, this contrasts with the validation of TAZ in WSS See response to Ref2, point2.

Our data appears to support activation of TAZ by WSS. We hope that the new draft of the manuscript does not misrepresent any changes we observe in TAZ signaling, and shows that our study has failed to find evidence for TAZ-dependent motility to date.

Advice: there is no need to stress the difference. The negative data on TAZ can be deleted. TAZ remains to be investigated.

We soften claims about the difference between YAP1 and TAZ throughout the text. We have not removed all the data on TAZ, however, as we believe it would be of interest to readers of Nature Communications. Knockdown

of TAZ suggests that WSS-induced increases in ANKRD1 and CTGF transcripts depend upon TAZ activation (Supplementary Fig. 4b). Using these same siRNAs against TAZ, we show that motility is not impaired (Fig. 3g). Consistent with the idea that TAZ does not impact taxis, rescue experiments with constitutively active and wild type TAZ (Supplementary Fig. 4a) further suggest that TAZ can not functionally compensate for YAP1 role(s) in motility.

TAZ was recently shown to be activated by shear stress in an independent report without data describing YAP1 response (Kim et al. 2014 below), thus it may be of interest to readers to understand that each of these transcriptional cofactors may have distinct functions in the context of biomechanical force, and specifically shear stress.

Kim KM, Choi YJ, Hwang J-H, Kim AR, Cho HJ, et al. (2014) Shear Stress Induced by an Interstitial Level of Slow Flow Increases the Osteogenic Differentiation of Mesenchymal Stem Cells through TAZ Activation. PLoS ONE 9(3): e92427

2) The authors are not doing the few, key experiments requested. I am referring to the YAP target genes (and lines 156-164). (point 2 ref 1 and points 1 and 4 of ref2). A) Ingenuity Knowledge Base can be at the most used to start, as a suggestion. But not taken as a fact. B) Validation is mandatory. As such, on my former point 4 "Examination of genes found previously by ChIP- Seq to be bound by YAP1 and those classically defined as YAP1/TAZ targets revealed" this remains a flawed statement. The authors should validate these genes by Chip-PCR in their own cells and conditions.

We now include qRT-PCR at various time points for the genes referenced in a new figure dedicated to independent validation: PPP1R3C, PPP2CB, PPP2CA, PPP2R3B, WWC1, SMAD3, YWHAB, YWHAG, and CYR61 (Supplementary Figure 2a). Other genes validated in Figure 2 include ANKRD1, CTGF, and AMOTL2. We also provide validation of select motility-related genes ADRB2, ETS1, MEF2C, PLAUR, and TGFB2 in Supplementary Figure 2b.

We have withdrawn statements asserting that differentially expressed genes are YAP1 targets. Specifically, we indicate “...expression of these genes was validated at several time points following initiation of WSS” and have removed the statement “classically defined as YAP1/TAZ targets”. We avoid any statement regarding transcriptional “targets” or direct regulation by YAP1 with respect to our own data. The terms “well-characterized” and “well-defined” were removed throughout the text and figure legends to de-emphasize claims that genes discussed are bona fide YAP1 targets.

ChIP-PCR is not technically feasible with the microfluidic platform described in the current manuscript. We are currently working toward development of a new system capable of producing sufficient material for the number of cells required for this analysis.

Alternatively, it would be easier to stay with obvious YAP/TAZ targets CTGF-CYR-ANKRD1 (and deleted the others) and on one or two of them show the differential recruitment of YAP by Chip-PCR after WSS.

qRT-PCR validation is now provided for ANKRD1, CTGF, CYR61, and AMOTL2 in Figure 2c and Supplementary Figure 2a. Unfortunately, we are unable to provide ChIP-PCR data with our current microfluidics platform.

We have replaced the heatmap in Figure 2b with one based exclusively on YAP1 targets validated by overlap in ChIP-seq promoter occupancy and loss or gain of function gene expression change in cardiomyocytes and glioblastoma cells (Lin et al. 2015 and Stein et al. 2015). This focused list of genes presented in Supplementary Table 2 represents a more informative tool for assessment of the expression change we see with WSS (of 83 YAP1 targets, 17 are upregulated and 3 downregulated). Paired with new qRT-PCR validation (Supplementary Fig. 2), we intend to convey that there is a general change in YAP1 signaling, de-emphasize any direct regulation by YAP1, and provide additional experimental evidence to support global transcriptome data as our entry point.

I have no idea what Dicer means, for example (see also Fig6, where actual checking of Dicer levels by qRT-PCR (same in WSS with and without YAP) do not validate the microarray (6a)). And neither Dicer nor others of the genes

listed in 6a fits the notion of established YAP/TAZ targets.

Dicer1 qRT-PCR data is removed; upon careful examination of the sources for our original gene list, Dicer1 emerged from a Hippo interactome study but has not been shown to be a bona fide YAP1 target gene.

We de-emphasize claims that differentially expressed genes are YAP1 targets through textual improvements and by moving Figure 6a and 6b (siYAP microarray and qRT-PCR data) to supplementary figures.

In other words: I do appreciate the YAP hype on cell mechanics and Hippo signaling and I am fully in favor of data driven hypotheses if they really to end up on YAP. But they are really stretching this approach to the limit; it seems to me "data driven with a bias" (= canalizing results toward the obvious YAP culprit). For example, it would be nice, if possible, to offer a revalidation by searching an overlap between bona fide YAP targets taken from the lists of direct targets that I saw recently published by a few groups (including Novartis and F. Camargo's labs).

Several HIPPO factors, as defined by IPA, have now been validated by qRT-PCR as differentially expressed and were responsible for significant change in this IPA canonical pathway with WSS ($p=0.02$). Secondary analysis of recently described YAP1 target genes (Stein et al. 2015, Lin et al. 2015) has now also been incorporated into Figure 2b based upon the work described by reviewer 2 above. We also re-evaluated our gene expression changes with genes shown to be directly regulated by YAP1/TAZ/TEAD enhancers (work by S. Piccolo referenced in the first review – Zanconato et al. 2015) and present those updates in the text, Supplementary Table 3, and Supplementary Figure 10a.

Similarly, the phrase copied on my former point 1: "Notably, components of the HIPPO signaling pathway were differentially expressed, including several phosphatases (PPP1R3C, PPP2CB, PPP2CA, and PPP2R3B), transcription regulators (WWC1 and SMAD3), and signal transduction factors (YWHAB, YWHAE, and YWHAG)" (and the title of their paragraph, see 139) continues to make absolutely no sense to me. This can only be deleted, in

absence of experimental validations.

We now provide qRT-PCR validation of the expression of these genes (Supplementary Fig. 2). We state that “Altered expression of genes involved in HIPPO signaling prompted us to conduct an in silico examination of YAP1 target genes...”. We also modify the title of this results section to draw attention away from HIPPO signaling and state “YAP1 signaling is altered by WSS”.

3) Rev.1 asked about potential integrin connection (her/his point 3) and see my own Rev2 point 3. I was not convinced about the Y127 phosphorylation (a LATS target residue) or LATS data, and I am even less convinced now. I had not asked for any experiment. I just suggested to attenuate and be more careful in the conclusions. Phospho-YAP as read-out is fine, but it is a downstream step of a series of events that they do not know (and do not need to address here). In contrast in this revision they add a western on LATS1 phosphorylation that complicates the message into an unexplored area, simply shifting the problem to an upper level (with its own issues) without really solving much.

We include new data showing that WSS activates focal adhesion kinase (FAK). FAK activation by integrin signaling is an established mechanism that correlates with prostate cancer migration and metastasis. We chose to consult with Sue-Hwa Lin and Gary Gallick at MD Anderson to select the most appropriate methodologies to interrogate a potential dependence upon FAK for YAP1 regulation, as they have extensive experience with integrin, FAK, and Src signaling in prostate cancer. Based upon these recommendations, we show that inhibition of FAK does impact overall levels of YAP1 nuclear localization and YAP1 phosphorylation in both static and WSS cultures (Fig. 4h-j). Notably, effects of FAK inhibition are modest in WSS cultures, suggesting that WSS does not require FAK to exert its effects on YAP1.

LATS data from the first revision has been removed.

A) the Ab is not validated by siRNAs against LATS1. Blots are not normalized to total LATS1. And LATS2?

We have begun validating antibodies for LATS1 and LATS2 through siRNA and all appropriate controls. However, as reviewer 2 has suggested, this requires an extensive amount of work and will likely constitute another paper.

B) Even if data of point A were present, the data would remain purely correlative, without making a pathway, as the authors are instead suggesting in the diagram of Fig.6.

Direct regulation of LATS by WSS is now removed, with instead a model that reflects the prevailing view that cytoskeletal tension poises the cell to be receptive to inhibition by HIPPO signaling. The modification to the model presented in Figure 6 includes an independent pathway of regulation by LATS through an unknown kinase “?”. In the legend for the model presented in Figure 6, we state “Although our data is consistent with LATS as a contributor to regulation, we cannot exclude alternate scenarios that include other kinases that post-transcriptionally modify YAP1.”

C) bringing UP LATS phosphorylation means that we need to bring up regulation on its own kinase, that is (mainly) MST etc. All untested steps and regulations. Something for a different and dedicated paper.

We agree that careful examination of the kinase cascade could constitute an entirely new paper and lies outside the scope of the current manuscript.

D) the authors do not know the function of LATS1, the role of LATS2, nor the functional role of YAP phosphorylation. There is no causality between these inferred biochemical interactions and the biology here investigated. Certainly there are no connections with cofilin etc. There is no proof that LATS mediate the effects of Rock-cofilin.

LATS data is removed and its role is de-emphasized throughout the text.

E) Although the rescue of LATS-phosphorylation by Rock inhibitor is interesting, the data does not discriminate between a direct regulation, a parallel event or a feedback. As such, the diagram of Figure 6 remains

flawed.

We have modified the model presented in Figure 6 to add a parallel pathway that includes an unknown kinase regulating LATS1/2, detailed above.

F) I suggest to delete the new data on LATS1 and to focus on the regulation of ROCK by integrin or other contextual mechanical signals (Shape, rigidity etc).

Agree, we have now removed this data and focus instead on FAK as a downstream effector of integrin signaling.

In sum: this remains a potentially interesting paper for Nat. Comm. However, they should offer a new revision in which they focus and word carefully their conclusions, remove unnecessary and confounding data, and provide clear validation of their data-driven entry point.

Reviewer #3 (Remarks to the Author):

In this manuscript, the authors identify a role for YAP1 in promoting cancer cell motility at shear stress levels representative of flow in lymphatic vessels, but not blood vessels. To conduct these experiments, the authors used a microfluidics platform of biologically relevant stiffness (~1 MPa). Overall, the authors have performed a thorough and comprehensive study supporting their hypothesis that YAP1, but not TAZ, is important to shear stress-mediated cancer cell motility. This response is mediated by ROCK, LIMK and YAP1-TEAD interactions. The authors are addressing an important but not well-studied aspect of cancer cell metastasis. Metastasis via the lymphatics is an important process in many cancers, but there are numerous unanswered questions about the biology of this process. The authors shed some light on one aspect of this process. In the revised manuscript, the authors have included additional control data that address my concerns and have strengthened the manuscript.

Reviewer #2 (Remarks to the Author)

The revised MS addressed all my concerns and I think it is now suitable for publication.